# Plan, Decouple, Assimilate:
# Physics-Aware Object Insertion in Remote Sensing Imagery

**Yingyan Hou** [1 2 3 4]  **Xianchi Dong** [1 2 3 4]  **Chao Ren** [1 4]  **Wanxuan Lu** [1 4]  **Zihan Wei** [1 2 3 4]  **Hongfeng Yu** [1 4]
**Yixiao Wang** [1 4]  **Xian Sun** [1 2 3 4 †]

## Abstract

Object insertion has emerged as a promising augmentation paradigm for the label scarcity and long-tailed distributions in remote sensing, generating training samples by synthesizing target instances onto real backgrounds. However, existing methods suffer from three critical issues: (i) semantic placement inconsistency, (ii) radiometric inconsistency with illumination and atmospheric conditions, and (iii) textural discontinuity. To address these, we propose a physics-aware method, "Plan, Decouple, Assimilate" (PDA), for generating high-fidelity training samples. In the planning stage, the Planning (P) module automatically generates geometrically valid bounding boxes. In the generation stage, a dual-module design synthesizes the target instance: the Decoupling (D) module employs Asymmetric Spectral Adaptation to disentangle structural identity from environmental illumination, while the Assimilation (A) module uses Neighborhood-Aware Texture Assimilation to harmonize the local manifold. By integrating these modules, PDA enforces multi-level consistency from global geometry to local micro-textures. Extensive experiments verify that PDA outperforms state-of-the-art methods in generative quality, reducing whole-image FID by **15.7%** over the strongest baseline, and substantially improves downstream detection, boosting average mAP50 by **+17.07** points over the real data.

---
[1]Aerospace Information Research Institute, Chinese Academy of Sciences, Beijing 100094, China [2]University of Chinese Academy of Sciences, Beijing 100190, China [3]School of Electronic, Electrical and Communication Engineering, University of Chinese Academy of Sciences, Beijing 100190, China [4]National Key Laboratory of Target Cognition and Application Technology (TCAT), Aerospace Information Research Institute, Chinese Academy of Sciences, Beijing 100094, China. Correspondence to: Xian Sun <sunxian@aircas.ac.cn>.

*Proceedings of the $43^{rd}$ International Conference on Machine Learning*, Seoul, South Korea. PMLR 306, 2026. Copyright 2026 by the author(s).

## 1. Introduction

Remote sensing object detection has been widely used in a range of applications, such as maritime surveillance (Rekavandi et al., 2025), airport traffic monitoring (Zhou et al., 2021), and strategic facility analysis (Bandarupally et al., 2020). It aims to facilitate the extraction of high-level semantic information from the earth observation data, serving as the foundational layer for automated geospatial analytics (Gui et al., 2024). However, the efficacy of deep detectors is increasingly hampered by a long-tailed distribution (Gao et al., 2024) and exhibits notable vulnerability to data-induced perturbations (Mei et al., 2024). Unlike natural imagery, collecting samples for specialized targets (e.g., uncommon aircraft) is often obstructed by security restrictions and prohibitive acquisition costs, leading to a severe scarcity of well-annotated datasets for rare categories (Wang et al., 2022).

To mitigate this data bottleneck, object insertion methods that synthesize target objects into authentic backgrounds have emerged as a promising paradigm for enriching training samples (Yang et al., 2023; Liu et al., 2025). Recent object insertion methods predominantly leverage diffusion-based generative models to synthesize target objects into real backgrounds (Chen et al., 2024b; Tang et al., 2025; Song et al., 2023; Hu et al., 2026). While these approaches have achieved visual fidelity, their utility for downstream tasks remains compromised due to three critical issues, as shown in Figure 1. First, existing methods often yield semantically inconsistent placements (e.g., aircraft in forests), violating the fundamental geographical context of remote sensing scenes. Second, generative models often overlook the underlying imaging geometry, synthesizing objects with shadow orientations and shading profiles that contradict the scene's solar azimuth and atmospheric conditions, thereby violating radiometric consistency. Third, textural discontinuity leads to visible boundary artifacts, colloquially known as the sticker effect. This phenomenon arises because the inserted object's inherent micro-texture fails to match the background's granularity and noise statistics, causing the object to appear pasted rather than naturally imaged.

To jointly address the above issues, we propose the "Plan,

**(a) Semantic Placement Inconsistency**

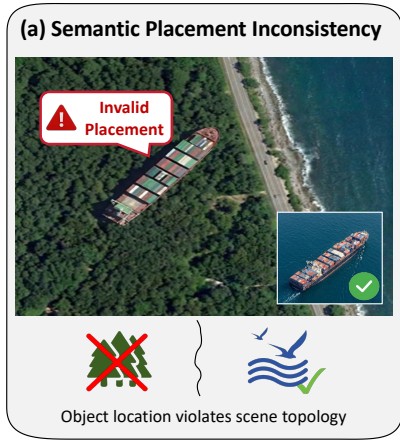

Object location violates scene topology

**(b) Radiometric Inconsistency**

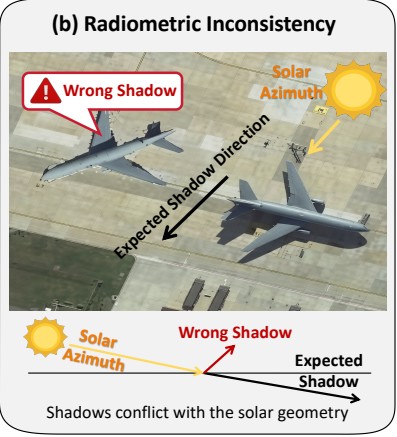

Shadows conflict with the solar geometry

**(c) Textural Discontinuity**

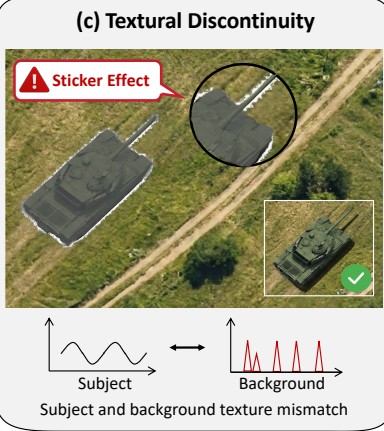

Subject and background texture mismatch

*Figure 1.* Illustration of three critical issues in remote sensing object insertion. (a) Semantic Placement Inconsistency: objects placed in geographically implausible regions, violating scene context; (b) Radiometric Inconsistency: misaligned shadow orientations and shading profiles that contradict the scene's solar azimuth; (c) Textural Discontinuity: the subject texture does not match the background texture.

Decouple, Assimilate" (PDA) method with a hierarchical framework that reformulates object insertion as a physics-aware generative process in remote sensing imagery. Unlike existing approaches, our proposed PDA method can explicitly orchestrate the interaction between objects and their corresponding environments. By strictly enforcing solar, atmospheric, and textural coherence, our PDA produces physically realistic synthetic data that effectively boosts downstream detection performance. We design three synergistic modules, including a Planner (P) module, a Decoupling (D) module, and an Assimilation (A) module, to cope with three critical issues. These three modules are mutually supportive, and together form a complete hierarchical framework. The main contributions are summarized as follows:

- Our proposed PDA method reformulates object insertion via strategically integrating systematic modules to generate high-fidelity training samples for remote sensing object detection. This physics-aware way can enforce multi-level consistency spanning global geometry to local micro-textures.

- We design a Physics-Grounded Scene Layout Planner (P) that leverages geometric distance fields to automatically generate topologically valid bounding boxes. We further design an Asymmetric Spectral Adaptation (ASA) mechanism that Decouples (D) structure from illumination by synergizing Frequency-Aware Decoupling with Implicit Solar Context Regulation, achieving photorealistic lighting without corrupting geometric details. Finally, we design a Neighborhood-Aware Texture Assimilation (NATA) (A) that guides inserted objects to inherit the granular statistics of their immediate surroundings, ensuring seamless integration and eliminating the sticker effect.

- Extensive experiments on multiple remote sensing

benchmarks demonstrate that our proposed PDA method achieves state-of-the-art performance reducing FID by 15.7% of whole image and improve the PSNR by 15.5% of insert image in the image generation over the strongest baseline, significantly improving downstream object detection accuracy, achieving mAP50 up to 80.26%.

## 2. Related Work

**Generative Models** Generative models have witnessed a paradigm shift from generative adversarial networks (GANs) (Goodfellow et al., 2014) to denoising diffusion probabilistic models (Ho et al., 2020). While GANs were previously dominant, they often suffer from training instability and mode collapse. In contrast, diffusion models, which generate data by reversing a stochastic noising process, have achieved satisfactory performance in high-fidelity image synthesis (Dhariwal & Nichol, 2021). To enhance manageability, latent diffusion models (LDMs) (Rombach et al., 2022) shifted the diffusion process to a lower-dimensional latent space, significantly reducing computational costs while maintaining visual quality. Furthermore, parameter-efficient fine-tuning strategies like Low-Rank Adaptation (LoRA) (Hu et al., 2022) have empowered these models with adaptability to specific styles or concepts without retraining the entire backbone. These advancements serve as the backbone of our framework.

**Object Insertion and Image Editing** Object insertion aims to seamlessly place foreground objects into background scenes. Early methods (e.g., copy-paste, Poisson blending) (Pérez et al., 2003) often ignore semantic context, while diffusion models (Rombach et al., 2022) enable reference-instruction-guided insertion with spatial controls (e.g., boxes or depth) (Zhang et al., 2023; Li et al., 2023),

as demonstrated by AnyDoor (Chen et al., 2024b), MimicBrush (Chen et al., 2024a), ACE++ (Mao et al., 2025), OmniPaint (Yu et al., 2025), Insert Anything (Song et al., 2025), UniCombine (Wang et al., 2025), Qwen Image Edit (Wu et al., 2025), and OminiControl (Tan et al., 2025). However, these general-purpose models are mainly trained on natural images (e.g., COCO) (Lin et al., 2014) and suffer from notable domain gaps in remote sensing (Liu et al., 2024), frequently producing inconsistent viewpoints or non-physical illumination, which calls for domain-specific adaptation to ensure physical realism.

**Generative Models in Remote Sensing** In the remote sensing domain, generative models have been increasingly adopted for tasks such as super-resolution (Liu et al., 2022; Wang & Sun, 2025) and data augmentation (Sousa et al., 2025; Yuan et al., 2023). Recent foundation models, such as CRS-Diff (Tang et al., 2024), have extended Stable Diffusion by incorporating geospatial metadata to generate context-aware satellite imagery. Some works have also explored using segmentation maps to guide generation (Toker et al., 2024; Deng et al., 2025). Despite these advances, a critical gap remains in high-fidelity object insertion (Han et al., 2025). Existing methods typically follow a black-box paradigm that neglects the rigid opto-physical constraints of overhead imagery. They often fail to decouple illumination from structure, leading to shadow directionality mismatches (Zhang et al., 2025), and struggle to match the micro-texture statistics of specific terrains (Tsai et al., 2017; Cong et al., 2020). Moreover, intelligent layout planning remains underexplored, with most methods relying on manual placement (Khammari et al., 2024). Our framework addresses these limitations by integrating physics-grounded planning with spectral-latent dynamics on top of a controllable generative backbone.

## 3. Preliminaries

### 3.1. Problem Formulation

Given a remote sensing background image $I_{bg} \in \mathbb{R}^{H \times W \times 3}$ and a target object image $I_{sub} \in \mathbb{R}^{H \times W \times 3}$, our goal is to generate a composite image $I_{out} \in \mathbb{R}^{H \times W \times 3}$ where the object is inserted at a coherent location in background image, which can be decomposed into the following two coupled sub-problems: (i) *Geometric Planning*, first seek an optimal insertion bounding box (mask) $M \in \{0, 1\}^{H \times W}$, such that the placement maximizes semantic plausibility with respect to the background topology; (ii) *Physics-Aware Generation*, given $M$, $I_{bg}$, and $I_{sub}$, our aim is to synthesize the content of object $x_{fg}$ within the masked region, while satisfying this conditional distribution:

$$x_{fg} \sim p(x|I_{bg}, M, I_{sub}). \tag{1}$$

### 3.2. Subject-Driven Condition Injection

To integrate subject reference images into the DiT architecture, we follow OminiControl (Tan et al., 2025) and employ a Unified Sequence Processing strategy. Specifically, latent tokens from the reference image $C$ are concatenated with the noisy image tokens to form a joint sequence. Crucially, since subject-driven generation requires semantic consistency rather than strict spatial alignment, we implement a shifted position encoding mechanism. Unlike spatial control tasks that typically share coordinate systems, we deliberately isolate the condition tokens by applying a substantial fixed offset to their position indices. This global index translation creates a disjoint coordinate arrangement within the rotary embedding space, ensuring zero spatial overlap between the condition and generation tokens. Consequently, the attention mechanism is structurally precluded from overfitting to local spatial correspondences and is instead compelled to aggregate global semantic relationships for robust identity preservation.

## 4. Methodology

### 4.1. Overview

To systematically resolve the critical impediments of semantic placement inconsistency, radiometric inconsistency, and textural discontinuity in remote sensing object insertion, we propose the `PDA` method. Unlike monolithic generation approaches, `PDA` is grounded in the theoretical premise that a physically coherent insertion must satisfy constraints across three orthogonal dimensions. Formally, we decompose the intractable conditional probability $p(x|I_{bg}, I_{sub})$ into three disentangled priors, each mapping directly to a specific stage in our architecture. The detailed function can be formulated as:

$$p(x|I_{bg}, I_{sub}) \approx \underbrace{p(x_{\text{pos}}|\nabla\mathcal{D})}_{\text{Stage I: Plan (P)}} \cdot \underbrace{\overbrace{p(x_{\text{spec}}|c_{\text{solar}})}^{\text{Decouple (D)}} \cdot \overbrace{p(x_{\text{tex}}|\mathcal{N}(x))}^{\text{Assimilate (A)}}}_{\text{Stage II: Generation}}.$$

$$\tag{2}$$

Guided by this factorization, the whole procedure formalized in Algorithm 1 operates with two coordinated stages.

**Stage I-Planning**, `PDA` implements the Planner (`P`) prior $p(x_{\text{pos}}|\nabla\mathcal{D})$ to eliminate geometric invalidity. Instead of relying on heuristic placement, this stage functions as a topological constraint solver. By leveraging the Euclidean Distance Transform (EDT) to identify Ridge Anchors with maximal spatial clearance, it strictly confines the object's pose to the background's valid semantic manifold, ensuring physical admissibility before any pixel synthesis begins.

**Stage II-Generation**, `PDA` orchestrates the Decoupling (`D`) and Assimilation (`A`) priors through a unified flow matching backbone. To resolve radiometric inconsistency

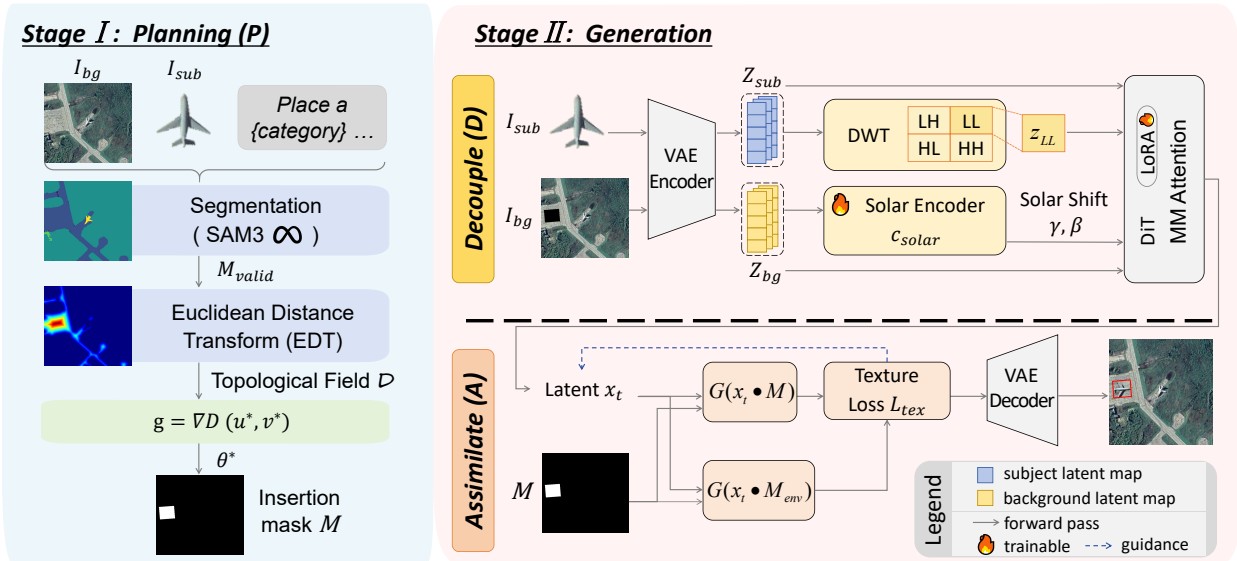

*Figure 2.* Overall pipeline of PDA with two stages: Stage I (Planning, P) predicts a valid insertion pose on the background, and Stage II (Generation) performs decoupling (D) and assimilation (A) to synthesize radiometrically and texturally coherent insertions.

$p(x_{\text{spec}}|c_{\text{solar}})$, the ASA-based decoupling module acts as a spectral filter that explicitly separates structural identity from environmental illumination, modulating low-frequency components to align with the scene's implicit solar context. Simultaneously, to mitigate texture artifacts $p(x_{\text{tex}}|\mathcal{N}(x))$, the NATA-based assimilation module enforces local stationarity. By treating background noise as a stationary process and injecting gradient guidance, it ensures the statistical convergence of micro-textures between the generated object and its immediate neighborhood.

### 4.2. Physics-Grounded Scene Layout Planner (P)

**Semantic-Geometric Parsing** We first interpret the scene's functional areas. We leverage SegEarthOV3 (Li et al., 2026), a custom open-vocabulary segmenter based on SAM3 (Carion et al., 2025), to parse the background $I_{bg}$. We define a target-specific category set $\mathcal{C}_{category}$. The segmentation map $S$ is filtered to generate a binary Valid Region Mask $M_{valid}$:

$$M_{valid}(u,v) = \begin{cases} 1 & \text{if } S(u,v) \in \mathcal{C}_{category} \\ 0 & \text{otherwise} \end{cases}. \quad (3)$$

This mask strictly confines the placement to semantically permissible grounds.

**Topology-Aware Placement.** To determine the single optimal pose $\mathbf{p}^* = (u^*, v^*, \theta^*)$ within the valid placement region $M_{valid}$, we analyze the scene's geometric topology using EDT. First, we compute the distance field $\mathcal{D}$, which maps every spatial location $\mathbf{x} = (u,v) \in M_{valid}$ to its minimum Euclidean distance from the nearest invalid boundary

$\partial M_{valid}$. This is formally defined as:

$$\mathcal{D}(\mathbf{x}) = \inf_{\mathbf{b} \in \partial M_{valid}} \|\mathbf{x} - \mathbf{b}\|_2, \quad (4)$$

where $\mathbf{b}$ represents a point on the boundary set $\partial M_{valid}$, and $\|\cdot\|_2$ denotes the $L_2$ norm. In this field, high-value regions correspond to the topological medial axes of spatially extensive areas. Consequently, we identify the optimal spatial coordinates $(u^*, v^*)$ by solving for the global maximum of $\mathcal{D}$, ensuring the object is anchored at the location with maximal clearance:

$$(u^*, v^*) = \arg \max_{\mathbf{x} \in M_{valid}} \mathcal{D}(\mathbf{x}). \quad (5)$$

To further ensure the object complies with the infrastructure's semantic flow, we determine the optimal orientation $\theta^*$ via Gradient-Based Alignment. Let $\nabla$ denote the gradient operator; the local gradient vector $\mathbf{g} = \nabla \mathcal{D}(u^*, v^*) \in \mathbb{R}^2$ indicates the direction of steepest descent towards the nearest boundary. We constrain the object's orientation $\theta^*$ to be orthogonal to $\mathbf{g}$ (i.e., $\theta^* \perp \mathbf{g}$), guaranteeing that the inserted instance aligns parallel to the principal axis of the edge. The final output is the rigorous geometric pose $\mathbf{p}^*$ that satisfies both spatial optimality and semantic alignment.

### 4.3. Asymmetric Spectral Adaptation Decoupling (D)

**Frequency-Aware Decoupling** To separate environmental illumination from the object's geometry, we employ a dual-stream mechanism based on frequency decomposition. Since illumination and atmospheric properties primarily reside in the low-frequency domain, we first apply the Discrete Wavelet Transform (DWT) to the subject latent $z_{sub}$ to extract its low-frequency component $z_{LL}$. We input this

---

**Algorithm 1:** The proposed `PDA` method

---

**Input:** Background $I_{bg}$, Subject $I_{sub}$, Category Set
$\quad\quad \mathcal{C}_{category}$
**Output:** Composite Image $I_{final}$
    `// Stage I: Planning`

1   **Plan (P)** valid placement region $M_{valid}$ via semantic
    parsing and compute topological field
    $\mathcal{D} \leftarrow \text{EDT}(M_{valid})$

2   Identify optimal ridge anchor $\mathbf{p}^* = (u^*, v^*)$ by
    maximizing $\mathcal{D}$ and align orientation $\theta^*$ orthogonal to
    gradient $\nabla\mathcal{D}$

3   Construct insertion mask $M$ and dilation mask $M_{env}$
    based on computed pose $(u^*, v^*, \theta^*)$
    `// Stage II: Generation`

4   Encode inputs to latents $z_{bg}, z_{sub}$ and extract implicit
    solar context $\mathbf{c}_{solar}$ from masked background

5   Map $\mathbf{c}_{solar}$ to affine modulation parameters $\{\gamma, \beta\}$ via
    MLP $\phi(\cdot)$ and sample noise $x_1 \sim \mathcal{N}(0, \mathbf{I})$

6   **for** $t = 1 \ldots 0$ **do**

7      **Decouple (D)** illumination via **ASA** to modulate
       spectral features $\mathbf{V}_{mod}$ inside DiT:

8        $\mathbf{V}_{mod} \leftarrow \mathbf{V} \odot (1 + \gamma) + \beta$      `// Eq.6`

9        $v_\theta \leftarrow \text{DiT}(x_t, t, \mathbf{V}_{mod})$

10     **Assimilate (A)** local texture via **NATA** by evaluating
       discrepancy $\mathcal{L}_{tex}$:

11       $x_{t,obj} \leftarrow x_t \odot M; \quad x_{t,env} \leftarrow x_t \odot M_{env}$

12       $\mathcal{L}_{tex} \leftarrow \|G(x_{t,obj}) - G(x_{t,env})\|_2^2$
        `// Eq.8`

13     Rectify flow trajectory $\hat{v}_t$ using texture gradient and
       advance ODE solver:

14       $\hat{v}_t \leftarrow v_\theta - \lambda(t)\nabla_{x_t}\mathcal{L}_{tex}$      `// Eq.9`

15       $x_{t-dt} \leftarrow \text{ODESolver}(x_t, \hat{v}_t, dt)$

16   **end**

17   **return** *Decoded image* $I_{final} \leftarrow VAE.Dec(x_0)$

---

component into the LoRA, enabling the model to adjust global shading and color temperature without altering high-frequency structural edges. Simultaneously, to maintain the object's original shape, we feed the complete latent representation $z_{sub}$ into a parallel LoRA. This branch preserves the structural details, ensuring that the object's rigid geometry remains consistent even as the lighting conditions change.

**Implicit Solar Context Regulation** To ensure the generated object is spectrally consistent with the background, we introduce a regulation mechanism based on implicit environmental cues. First, we extract a global context vector $\mathbf{c}_{solar}$ directly from the masked background latent $z_{bg}$. Unlike methods dependent on explicit solar angles, our approach implicitly captures the scene's lighting atmosphere by aggregating high-level features from surrounding ground elements. Subsequently, we use this context to guide the generation process. We project $\mathbf{c}_{solar}$ to globally modulate the attention layers within the DiT backbone. This injection ensures that the entire attention mechanism is conditioned on the global environmental context. Formally, we map $\mathbf{c}_{solar}$ into affine scale ($\gamma$) and shift ($\beta$) parameters to mod-

ulate the Value matrix $\mathbf{V}$ in the DiT blocks:

$$\gamma, \beta = \phi(\mathbf{c}_{solar}), \quad \mathbf{V}_{mod} = \mathbf{V} \odot (1 + \gamma) + \beta, \quad (6)$$

where $\phi(\cdot)$ denotes a learnable projection network (MLP), $\mathbf{V} \in \mathbb{R}^{N \times D}$ represents the original Value features in the attention mechanism, $\mathbf{V}_{mod}$ denotes the modulated Value features, and $\odot$ indicates row-wise broadcasting multiplication. This operation aligns the feature distribution across the entire DiT with the background's implicit context.

### 4.4. Neighborhood-Aware Texture Assimilation (`A`)

**Local Neighborhood Definition** Texture consistency is a local property. A global alignment might introduce irrelevant textures from distant regions onto the object. Therefore, we define a local environmental region $M_{env}$ specifically for texture reference. Given the insertion mask $M$, which can be calculated from the pose $\mathbf{p}^* = (u^*, v^*, \theta^*)$ and the size of the subject, we compute a dilated mask $M_{dilated}$ via morphological dilation. This region $M_{env}$ is constructed by subtracting the original object mask $M$ from its morphologically dilated counterpart $M_{dilated}$. This spatial constraint ensures that the assimilation process is driven exclusively by the valid background pixels adjacent to the object boundaries, effectively filtering out global noise.

**Gram-Based Texture Guidance** To ensure seamless integration, we enforce texture consistency directly within the flow matching latent space, eliminating the need for computationally expensive backbone feature extraction. At each inference timestep $t$, let $x_t \in \mathbb{R}^{C \times h \times w}$ denote the noisy latent image. We define the texture assimilation objective as minimizing the statistical discrepancy between the generated object and its immediate surroundings within this latent domain. Specifically, we extract the foreground latent $x_{t,obj}$ and the environmental neighborhood latent $x_{t,env}$ using the binary mask $M$ and the annular neighborhood mask $M_{env}$:

$$x_{t,obj} = x_t \odot M, \quad x_{t,env} = x_t \odot M_{env}. \quad (7)$$

We leverage the Gram Matrix $G(\cdot)$ to capture stationary texture statistics independent of spatial semantics. The Texture Assimilation Loss $\mathcal{L}_{tex}$ is formulated as the squared Euclidean distance between the Gram matrices of the object and its local neighborhood:

$$\mathcal{L}_{tex} = \|G(x_{t,obj}) - G(x_{t,env})\|_2^2. \quad (8)$$

This formulation forces the generated latent codes to adopt the same spectral noise distribution and granularity as the surrounding background latents at the current noise level $t$.

**Gradient-Guided Flow Matching** To assimilate the texture, we employ energy-based guidance on the flow trajectory. Let $v_\theta(x_t, t)$ denote the velocity field predicted by the

flow matching model with parameters $\theta$ at time $t \in [0, 1]$. We introduce a texture consistency loss $\mathcal{L}_{tex}$ and compute its gradient with respect to the latent state $x_t$ to inject guidance into the ODE solver. Specifically, the guided velocity field $\hat{v}_t$ is formulated as:

$$\hat{v}_t = v_\theta(x_t, t) - \lambda(t) \cdot \nabla_{x_t} \mathcal{L}_{\text{tex}}(x_t), \tag{9}$$

where $\nabla x_t$ denotes the gradient operator with respect to the spatial coordinates of $x_t$, and $\lambda(t) > 0$ is a time-dependent scalar weighting factor. By correcting the velocity $v_\theta$ with this gradient term, the flow is steered to minimize the texture loss, thereby synthesizing high-frequency details that are statistically consistent with the background sensor noise.

### 4.5. Analysis

We analyze two properties of our proposed PDA that underpin geometric validity and texture assimilation.

**EDT Ridge Anchors Maximize Geometric Clearance.** Recall the EDT field $\mathcal{D}(\mathbf{x}) = \inf_{\mathbf{b} \in \partial M_{valid}} \|\mathbf{x} - \mathbf{b}\|_2$ and the ridge anchor $\mathbf{p}^* = \arg\max_{\mathbf{x} \in M_{valid}} \mathcal{D}(\mathbf{x})$. For placement safety, we use a conservative footprint bound: let $r$ be the radius of a disk that contains the object footprint after rotation (e.g., the circumradius of its rotated bounding box). If $\mathcal{D}(\mathbf{p}) \geq r$, then the disk $B(\mathbf{p}, r)$ lies entirely in $M_{valid}$ by the definition of $\mathcal{D}$, implying a collision-free placement under this bound. Moreover, selecting $\mathbf{p}^*$ maximizes $\mathcal{D}(\mathbf{p})$, hence maximizes the available clearance margin among all candidate locations in $M_{valid}$. This provides a principled rationale for geometric validity by construction.

**NATA as Energy-Based Flow Correction.** Recall the texture assimilation loss $\mathcal{L}_{tex}$ in Eq. (8) and guided velocity field $\hat{v}_t$ in Eq. (9). Under the continuous-time dynamics $\dot{x}_t = \hat{v}_t$, the chain rule yields as:

$$\frac{d}{dt} \mathcal{L}_{tex}(x_t) = \langle \nabla \mathcal{L}_{tex}(x_t), v_\theta(x_t, t) \rangle - \lambda(t) \|\nabla \mathcal{L}_{tex}(x_t)\|_2^2. \tag{10}$$

Eq. (10) shows that NATA introduces a negative quadratic term that counteracts texture inconsistency. In particular, whenever $\lambda(t) \|\nabla \mathcal{L}_{tex}\|_2^2$ dominates $\langle \nabla \mathcal{L}_{tex}, v_\theta \rangle$, the guidance enforces an instantaneous decrease of $\mathcal{L}_{tex}$, providing a sufficient condition for locally stationarizing micro-textures around the insertion boundary.

## 5. Experiments

### 5.1. Datasets and Implementation Details

**Datasets.** We construct our training and evaluation benchmarks by integrating two widely used remote sensing datasets: the FAIR1M subset (Sun et al., 2022) of SAMRS (Wang et al., 2023), and iSAID (Waqas Zamir et al., 2019),

which is derived from DOTA imagery (Xia et al., 2018). To ensure high visual fidelity, we additionally apply a data preprocessing pipeline leveraging instance segmentation masks and oriented bounding box (OBB) annotations. Small-scale instances are removed to avoid low-resolution artifacts. Valid target objects are extracted, centered on a $512 \times 512$ canvas, and processed with edge blurring to mitigate boundary aliasing. Meanwhile, corresponding objects in the original images are masked out to form background–location pairs. In total, we curate 19,163 training pairs (14,215 from SAMRS and 4,948 from iSAID), while an additional 1,718 pairs serve as a hold-out test set for evaluating generation quality.

**Implementation Details.** Our framework is implemented in PyTorch and fine-tuned on a single NVIDIA A100 GPU (80GB VRAM). The model is trained for 20,000 steps with a batch size of 4. Instead of conventional manual hyperparameter tuning, we adopt the Prodigy optimizer, an adaptive method that dynamically estimates step sizes. We set the initial learning rate to 1 to accelerate convergence while maintaining training stability. All input images are resized to $512 \times 512$ during both training and inference.

### 5.2. Generative Quality Assessment

**Comparative Analysis** To assess the visual realism and physical consistency of our synthesized samples, we compare PDA against representative diffusion-based frameworks in Table 1 and Figure 3. PDA achieves the lowest FID of 9.73 and the best Insertion Region scores, with a PSNR of 20.87 dB and an SSIM of 0.6249, surpassing the OminiControl baseline. These differences reflect distinct failure modes in the remote sensing domain. General-purpose inpainting methods such as AnyDoor, MimicBrush, Qwen Image Edit, and UniCombine lack priors for orthographic imaging geometry, and their outputs break down structurally. MimicBrush is the clearest case: it reaches a high Whole-Image SSIM of 0.93 mainly by copying the frozen background, yet its Insertion Region SSIM drops to 0.37, which shows it cannot generate valid objects. Controllable baselines such as Insert Anything and OminiControl retain the coarse geometric structure but show clear spectral inconsistencies. Insert Anything produces a global color deviation with unrealistic hue shifts, while OminiControl does not adapt to scene-specific illumination, so its synthesized objects look pasted in rather than naturally integrated. In contrast, PDA models the scene context explicitly through Solar-Guided Regulation, and its lighting and color distributions align most closely with the ground truth, consistent with its leading PSNR and LPIPS scores. Objects synthesized by PDA are therefore both geometrically faithful and radiometrically consistent with the scene. Additional qualitative results are in Appendix A.4.

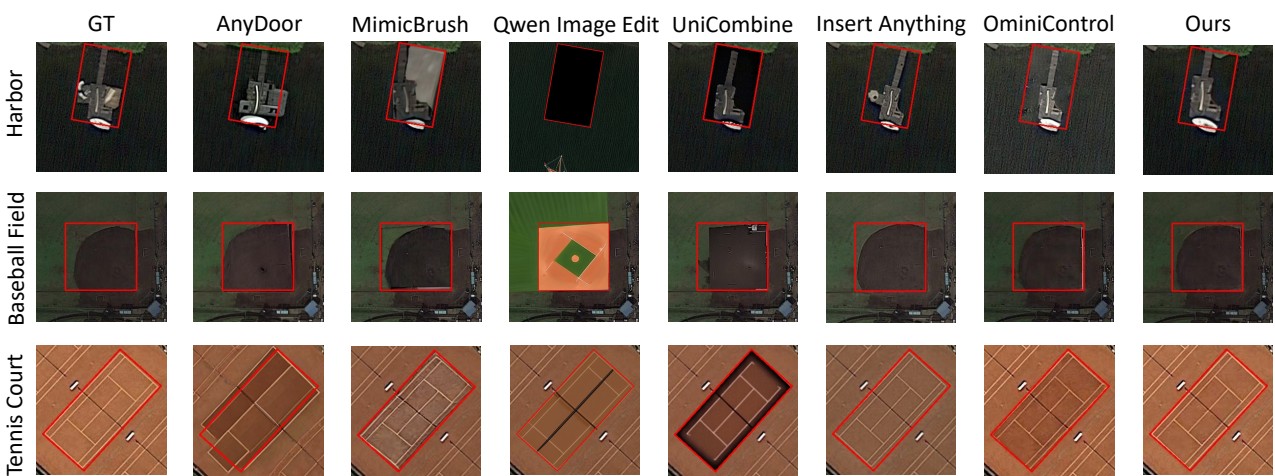

*Figure 3.* Qualitative comparison of object insertion. We compare our `PDA` with diffusion-based methods.

*Table 1.* Quantitative comparison with state-of-the-art methods for optical remote sensing object insertion. Whole-image metrics evaluate global visual fidelity, while insertion-region metrics more directly assess the realism and environmental compatibility of the synthesized object.

| Methods | Whole Image | | | | Insertion Region | | |
|---|---|---|---|---|---|---|---|
| | PSNR ↑ | SSIM ↑ | LPIPS ↓ | FID ↓ | PSNR ↑ | SSIM ↑ | LPIPS ↓ |
| AnyDoor (Chen et al., 2024b) | 24.24 | 0.8501 | 0.1040 | 21.47 | 15.70 | 0.4411 | 0.2897 |
| MimicBrush (Chen et al., 2024a) | 24.81 | **0.9337** | **0.0582** | 21.85 | 14.36 | 0.3676 | 0.3478 |
| Qwen Image Edit (Wu et al., 2025) | 18.41 | 0.7912 | 0.2044 | 46.08 | 10.72 | 0.2574 | 0.5698 |
| UniCombine (Wang et al., 2025) | 24.97 | 0.8869 | 0.0781 | 22.06 | 15.89 | 0.4566 | 0.2827 |
| ACE++ (Mao et al., 2025) | 17.44 | 0.3924 | 0.2659 | 29.24 | 15.89 | 0.4167 | 0.2757 |
| OmniPaint (Yu et al., 2025) | 24.32 | 0.8283 | 0.0871 | 18.14 | 16.83 | 0.4965 | 0.2156 |
| Insert Anything (Song et al., 2025) | 26.12 | 0.8893 | 0.0707 | 11.54 | 18.07 | 0.5463 | 0.1561 |
| OminiControl (Tan et al., 2025) | 25.22 | 0.8603 | 0.0839 | 12.05 | 17.90 | 0.5396 | 0.1669 |
| `PDA` (Ours) | **28.41** | 0.8901 | 0.0601 | **9.732** | **20.87** | **0.6249** | **0.1247** |

The best results are highlighted in **bold**, and the second-best are underlined.

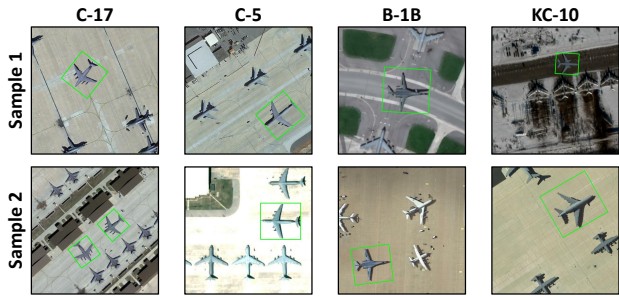

*Figure 4.* Case studies of synthetic data under different scenarios.

**Ablation Analysis**   To analyze the contribution of each component in `PDA`, we conduct an ablation study on the core generative modules, as reported in Table 2. The OminiControl baseline provides a solid starting point, with a Whole-Image PSNR of 25.22 dB, but it shows clear spectral inconsistencies, reflected in its relatively high FID of 12.05. Adding the NATA module yields moderate gains, improving the Insertion-Region SSIM from 0.5396 to 0.5441. This suggests that while NATA refines high-frequency micro-textures to match local terrain patterns, it is not sufficient on its own to close the global domain gap.

The largest improvement comes from the ASA module. With ASA enabled, the Whole-Image PSNR rises by 3.15 dB to 28.37 dB over the baseline, and the FID drops by 19% to 9.75. This indicates that spectral misalignment, rather than texture fidelity, is the dominant bottleneck in remote sensing object insertion. By explicitly harmonizing illumination and atmospheric characteristics, ASA makes the synthesized object radiometrically consistent with the background scene.

Finally, the full `PDA` framework, which integrates both modules, achieves the best overall performance, with a PSNR of 28.41 dB. This confirms that structural refinement and spectral fidelity contribute complementarily to the realism of the generated imagery.

*Table 2.* Ablation study on generative components. **Bold** indicates the best performance, and second best are underlined.

| Method Variants | Whole Image | | | | Insertion Region | | |
|---|---|---|---|---|---|---|---|
| | PSNR ↑ | SSIM ↑ | LPIPS ↓ | FID ↓ | PSNR ↑ | SSIM ↑ | LPIPS ↓ |
| Baseline (OminiControl) | 25.22 | 0.8603 | 0.0839 | 12.05 | 17.90 | 0.5396 | 0.1669 |
| + NATA (Gram) | 25.28 | 0.8607 | 0.0837 | 12.03 | 18.00 | 0.5441 | 0.1653 |
| + ASA (Solar+Fre) | 28.37 | 0.8898 | 0.0604 | 9.752 | 20.79 | 0.6218 | 0.1257 |
| PDA (Ours) | **28.41** | **0.8901** | **0.0601** | **9.732** | **20.87** | **0.6249** | **0.1247** |

### 5.3. Downstream Task Evaluation

**Performance of OBB Object Detection** To further verify the practical utility of our generated data, we construct a few-shot benchmark named MAR20-11-FewShot[1], derived from the MAR20 (Wenqi et al., 2023) dataset. We select 11 diverse categories and randomly sample 30 images per category, 330 in total, as the training set to simulate an extreme data-scarce scenario. We consider four augmentation strategies that expand the dataset by generating pseudo targets on the training set: Copy-Paste, CutMix (Burgert et al., 2025), OminiControl, and our proposed PDA[2]. Some samples generated by PDA are shown in Figure 7. We then evaluate how well each strategy improves oriented object detection under a zero-shot synthesis setting. Four representative OBB detectors are used for evaluation: Rotated Fast R-CNN (Ren et al., 2016), Oriented R-CNN (Xie et al., 2021), S2ANet (Han et al., 2021), and YOLO26 (Sapkota et al., 2025).

As shown in Table 3, detectors trained only on the 330 real images generalize poorly, with mAP50 ranging from 55.67% for S2ANet to 68.51% for Oriented R-CNN. Copy-Paste gives consistent gains, for example raising Rotated Fast R-CNN to 73.14%, which shows that increasing instance density helps when the pasted objects keep a realistic appearance. Its improvement is nonetheless limited by the low diversity of the original instances and by possible context inconsistency. OminiControl, by contrast, gives uneven results: it improves Oriented R-CNN but lowers Rotated Fast R-CNN from 58.13% to 55.22%, which suggests that its zero-shot samples can be visually plausible yet poorly aligned with the fine-grained category cues in MAR20-11. PDA delivers the strongest and most stable improvements across all detectors, with the largest gain on S2ANet, which rises by 21.69% to 77.36%, and it also pushes YOLO26 to 80.26%. We attribute these consistent gains to support-guided synthesis, which better preserves the class-specific texture and shape patterns from the few-shot references and produces synthetic targets that are more diverse and more

[1]Details about this benchmark are shown in Appendix A.2.
[2]The detailed workflow of Automated Multi-Object Dataset Synthesis is illustrated in Appendix A.1.

*Table 3.* Downstream oriented object detection performance (mAP50 in %) on the MAR20-11-FewShot benchmark under zero-shot optical synthesis. CP, CM, and OC denote Copy-Paste, Cut-Mix (Burgert et al., 2025), and OminiControl, respectively. The best results are in **bold** and the second-best are underlined.

| Detector | Real | +CP | +CM | +OC | PDA |
|---|---|---|---|---|---|
| Rotated R-CNN | 58.13 | 73.14 | 68.24 | 55.22 | **77.88** |
| Oriented R-CNN | 68.51 | 78.01 | 71.07 | 70.04 | **78.59** |
| S2ANet | 55.67 | 71.86 | 62.77 | 62.36 | **77.36** |
| YOLO26 | 63.51 | 75.54 | 76.79 | 70.01 | **80.26** |

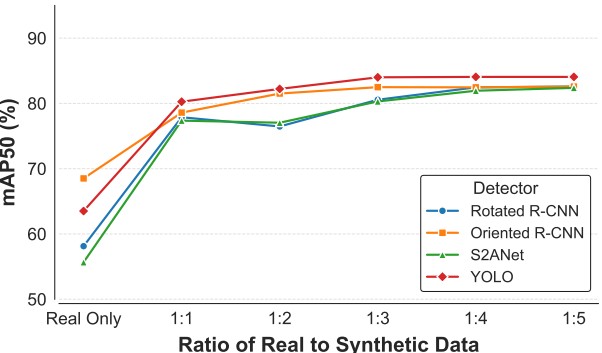

*Figure 5.* Effect of synthetic data ratio on detection performance.

discriminative for fine-grained OBB detection.

**Impact of Synthetic Data Ratio** To investigate the scalability and distributional robustness of our framework, we conduct a sensitivity analysis by varying the ratio of real to synthetic data from 1:1 to 1:5. As illustrated in Figure 5, the introduction of synthetic data yields consistent performance gains across all detector architectures. The most substantial improvement is observed at the initial 1:1 mixture, where Rotated Fast R-CNN and S2ANet achieve immediate boosts of +19.75% and +21.69%, respectively, effectively mitigating the few-shot bottleneck.

Performance continues to improve as the synthetic proportion increases, with YOLO26 reaching a peak of 84.06% at a 1:4 ratio. Although a saturation plateau emerges thereafter, suggesting a ceiling in discriminative feature extraction.

*Table 4.* Quantitative validation of physical consistency on the optical evaluation set. The shadow metrics evaluate the consistency of shadow overlap, intensity, orientation, and extent, while the radiometric metrics measure the alignment between the inserted object region and the surrounding background in terms of color statistics, brightness, color temperature, and intensity distribution.

| Method Variant | Shadow Consistency | | | | Radiometric Consistency | | | |
|---|---|---|---|---|---|---|---|---|
| | Shadow IoU ↑ | Depth ↓ | Dir. ↓ | Area ↓ | RGB ↓ | Bright. ↓ | Temp. ↓ | KL ↓ |
| Baseline (OminiControl) | 0.6489 | 0.0683 | 8.620 | 0.0735 | 14.37 | 14.34 | 0.0363 | 1.1944 |
| + NATA | 0.6510 | 0.0673 | 8.474 | 0.0740 | 14.25 | 14.24 | 0.0355 | 1.1766 |
| + ASA | 0.7533 | 0.0384 | 5.250 | 0.0445 | 3.593 | 3.555 | 0.0145 | 0.2483 |
| PDA (Ours) | **0.7550** | **0.0376** | **5.218** | **0.0437** | **3.587** | **3.549** | **0.0143** | **0.2451** |

The best results are highlighted in **bold**, and the second-best are underlined.

*Table 5.* Inference efficiency of different optical insertion variants on an NVIDIA A100 GPU. Time and memory are measured per image.

| Method Variant | Time (s/img) ↓ | Mem (GiB) ↓ | FID ↓ |
|---|---|---|---|
| Baseline (OminiControl) | 14.5 | 33.51 | 12.05 |
| + NATA | 14.9 | 33.91 | 12.03 |
| + ASA | 17.8 | 35.89 | 9.752 |
| PDA (Ours) | **18.2** | **36.30** | **9.732** |

*Table 6.* Quantitative validation of recursive multi-object synthesis on MAR20 under the zero-shot optical setting. To directly measure local insertion fidelity, all metrics are computed on the insertion region over 1,000 test cases.

| Scenario | PSNR ↑ | SSIM ↑ | LPIPS ↓ |
|---|---|---|---|
| Single insertion | 20.34 | 0.6676 | 0.1109 |
| Multi insertion (Recursive) | 19.23 | 0.6464 | 0.1312 |

Crucially, no performance degradation is observed even at the 1:5 ratio. This stability confirms that PDA generates samples that are distributionally compatible with real data, avoiding the domain shift or noise accumulation often associated with excessive synthetic augmentation.

### 5.4. Further Analysis

**Physical Consistency Validation.** To directly validate the physical correctness of synthesized objects, we measure both *geometric* (shadow consistency with ground truth) and *radiometric* (illumination/color agreement with the surrounding background) statistics on the iSAID evaluation set. As shown in Table 4, ASA reduces the shadow direction error from 8.62° to 5.25°, suppresses the brightness gap by 75%, and reduces the KL divergence by 79%, confirming that the implicit context $c_{solar}$ captures meaningful illumination structure rather than serving as an arbitrary feature.

**Multi-Object Recursive Insertion.** For multi-object scenes, the Planner first runs a Batch Layout phase with a dynamic occupancy mask $M_{occ}$ that guarantees collision-free placement, after which objects are inserted recursively: the background state is updated via Eq. (11), so that $c_{solar}$ is re-extracted from a scene containing all previously placed objects, which keeps the illumination self-consistent across the sequence. On 1,000 MAR20 cases, recursive insertion reduces PSNR by only 1.11 dB and SSIM by 0.021 relative to single insertion (Table 6), confirming that the design scales to many objects without accumulating artifacts.

**Computational Cost.** PDA generates one $512 \times 512$ sample in 18.2 s on a single A100, with 36.3 GiB peak memory. This is comparable to OminiControl at 14.5 s and faster than Insert Anything at 21.7 s, while PDA still attains the lowest FID. Its training cost is $1.14 \times$ that of OminiControl fine-tuning. The Planner runs once per background and its cost is fully amortized (Appendix A.5).

**Cross-Modality and Cross-Sensor Generalization.** The modular design makes PDA largely independent of the input modality. Without any architectural change, PDA fine-tuned on HRSID (Sentinel-1 SAR) outperforms OminiControl on all generative metrics, and its zero-shot transfer to SSDD ship detection achieves the best mAP50 across four detectors; for YOLO26, for example, mAP50 rises from 79.62 to 84.28. For amorphous targets at moderate resolution, PDA fine-tuned on WHU-OPT-SAR (5 m, forest) keeps a consistent margin over the baselines (Appendix A.6).

## 6. Conclusion

We proposed a physics-aware "Plan, Decouple, Assimilate" framework for remote sensing object insertion. Our PDA enforces multi-level consistency in geometry, illumination, and texture, enabling high-fidelity sample synthesis. Experiments on multiple remote sensing benchmarks confirm both superior generation fidelity and downstream utility, improving average mAP50 by 17.07 over the real-data baseline in downstream object detection. The results demonstrate that physics-aware insertion offers a practical solution to long-tailed data scarcity in remote sensing object detection.

## Impact Statement

This work contributes to remote sensing object detection by reducing data scarcity through physics-aware synthetic sample generation. We do not identify specific societal or ethical impacts beyond those already recognized for remote sensing technologies.

## Acknowledgements

This work was supported by the National Natural Science Foundation of China under Grant 62425115.

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

# A. Appendix

## A.1. Automated Multi-Object Dataset Synthesis

This subsection details the automated *Progressive Scene Population* pipeline that underlies the multi-object recursive insertion in Section 5.3 of the main paper, and which we use to construct the large-scale **MAR20-11-FewShot** benchmark. Unlike single-object insertion, this pipeline must handle inter-object spatial conflicts and maintain coherent context accumulation across sequential insertions. As outlined in Algorithm 2, the process operates in two distinct phases: Global Layout Planning and Recursive Generative Injection.

**Phase I: Batch Layout Planning (Global Topology Analysis).**   Instead of planning a single location, the goal of this phase is to identify a set of $N$ optimal, non-overlapping poses $\mathcal{Q} = \{\mathbf{p}_1, \ldots, \mathbf{p}_N\}$. We first compute the global Euclidean Distance Transform (EDT) field $\mathcal{D}$ on the valid semantic regions of the raw background $I_{bg}^{raw}$. To ensure spatial diversity and prevent collisions, we maintain a dynamic binary occupancy mask $M_{occ}$ (initialized as zeros) that tracks the utilized space. The planning proceeds iteratively:

1. Select the *ridge anchor* $\mathbf{p}_k = (u_k, v_k)$ by finding the global maximum of $\mathcal{D}$ within the unoccupied region (i.e., masking $\mathcal{D}$ with $1 - M_{occ}$).

2. Compute the optimal orientation $\theta_k$ orthogonal to the local gradient $\nabla\mathcal{D}(\mathbf{p}_k)$.

3. Mark the chosen location as occupied by updating $M_{occ}$: the region corresponding to the object's footprint at $\mathbf{p}_k$ is set to $1$ ($M_{occ}(p) \leftarrow 1, \forall p \in \text{Box}(\mathbf{p}_k)$), preventing subsequent instances from overlapping this area.

This iterative procedure yields a queued layout $\mathcal{Q}$ in which every planned pose is both geometrically valid and sufficiently clear of other instances.

**Phase II: Recursive Generative Injection.**   Since the generative model is conditioned on the immediate context, inserting all objects simultaneously would lead to inconsistent lighting and texture across instances. We therefore adopt a *recursive* injection strategy. Let $I_{curr}^{(k)}$ denote the background state at step $k$, initialized as $I_{curr}^{(0)} = I_{bg}^{raw}$. For each planned pose $\mathbf{p}_k \in \mathcal{Q}$:

1. **State preparation.** Generate a binary insertion mask $M_k$ at pose $\mathbf{p}_k$ and prepare a contextual canvas $I_{ctx} = I_{curr}^{(k-1)} \odot (1 - M_k)$.

2. **Physics-aware generation.** The generator synthesizes the object $x_{gen}$ conditioned on $I_{ctx}$ and the rotated subject $I_{sub}^{rot} = \text{Rotate}(I_{sub}, \theta_k)$. The Neighborhood-Aware Texture Assimilation (NATA) module then refines the output to $x_{refined}$, harmonizing it with the textures of both the original background and previously inserted objects.

3. **State update.** The background is explicitly updated to serve as the context for the next iteration:

$$I_{curr}^{(k)} \leftarrow I_{curr}^{(k-1)} \odot (1 - M_k) + x_{refined} \odot M_k. \tag{11}$$

The key property of this design is that each subsequent object is generated conditioned on a scene that already contains all previously placed objects, so the implicit solar context $\mathbf{c}_{solar}$ extracted from $I_{curr}^{(k)}$ reflects the evolving scene rather than the initial empty layout. Lighting and texture coherence are therefore preserved across arbitrarily many insertions, as quantitatively verified in Table 6 of the main paper.

## A.2. Construction of MAR20-11-FewShot Benchmark

To evaluate the efficacy of our method in data-scarce scenarios involving high-value strategic targets, we curate a specialized subset derived from the fine-grained MAR20 dataset (Wenqi et al., 2023). The original dataset contains 20 aircraft categories (labeled A1–A20). In our benchmark, we strategically exclude common fighter jets and smaller tactical aircraft to focus on Large-Scale Strategic Airframes, which present greater challenges for geometric consistency and texture assimilation during insertion.

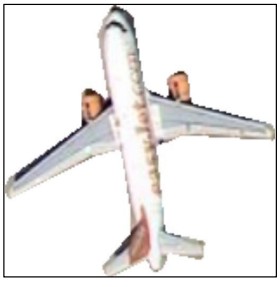 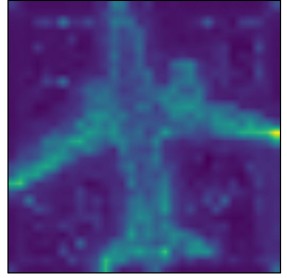 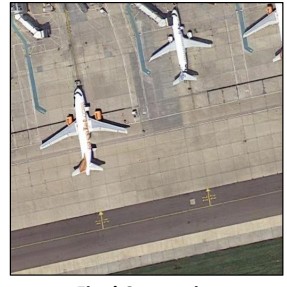 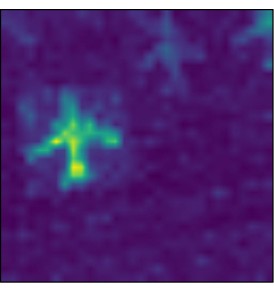

| **Reference Subject** | **Subject Self-attention** | **Final Composite** | **Cross-Attention** |

*Figure 6.* Attention map in `PDA`.

Specifically, we exclude the following 9 categories: A1 (SU-35), A5 (F-16), A12 (TU-22), A13 (F-15), A15 (F-22), A16 (FA-18), A17 (TU-95), A19 (SU-34), and A20 (SU-24).Consequently, we select the remaining 11 categories to construct the MAR20-11-FewShot benchmark. These categories encompass strategic transport aircraft, heavy bombers, and aerial refueling tankers, specifically: A2 (C-130), A3 (C-17), A4 (C-5), A6 (TU-160), A7 (E-3), A8 (B-52), A9 (P-3C), A10 (B-1B), A11 (E-8), A14 (KC-135), and A18 (KC-10). The detailed composition is listed in Table 7. For the few-shot setting, we randomly sample $K = 30$ images per category to form the support set. The detailed composition and their relative length scales are listed in Table 7, which helps us determine the size of the bbox.

*Table 7.* The composition of the MAR20-11-FewShot Benchmark. We select 11 large-scale strategic categories. Rel. Scale denotes the relative length ratio with respect to the fighter jet A5 (F-16) (length $\approx$ 15m) set as 1.00.

| ID | Code | Class Name | Rel. Scale | ID | Code | Class Name | Rel. Scale |
|----|------|------------|------------|----|------|------------|------------|
| 1 | A2 | **C-130** Hercules | 1.98 | 7 | A9 | **P-3C** Orion | 2.36 |
| 2 | A3 | **C-17** Globemaster III | 3.52 | 8 | A10 | **B-1B** Lancer | 2.95 |
| 3 | A4 | **C-5** Galaxy | 5.00 | 9 | A11 | **E-8** JSTARS | 3.09 |
| 4 | A6 | **TU-160** Blackjack | 3.59 | 10 | A14 | **KC-135** Stratotanker | 2.76 |
| 5 | A7 | **E-3** Sentry | 3.09 | 11 | A18 | **KC-10** Extender | 3.67 |
| 6 | A8 | **B-52** Stratofortress | 3.22 | | | | |

## A.3. Implementation Details of Baseline Methods

To rigorously evaluate the performance of `PDA`, we compare it against eight representative state-of-the-art object insertion and editing frameworks. AnyDoor (Chen et al., 2024b): A diffusion-based object customization method that employs a High-Frequency Filter and a Teleporting Module. It aims to transfer high-fidelity details from the reference object to the target location for zero-shot synthesis. MimicBrush (Chen et al., 2024a): A zero-shot editing framework focusing on texture transfer. It utilizes a Reference U-Net to extract semantic textures from the subject and injects them into the main generation process via self-attention mechanisms to imitate the reference appearance. Qwen Image Edit (Wu et al., 2025): A large-scale multimodal framework tailored for instruction-based image editing. It leverages the reasoning capabilities of Qwen-VL to interpret complex natural language commands and perform corresponding edits, representing the state-of-the-art in text-guided general image manipulation. UniCombine (Wang et al., 2025): A unified multi-conditional generation framework. It integrates diverse control signals (e.g., text, edges, reference images) into a shared embedding space, allowing for comprehensive control over the generation process through a unified transformer backbone. ACE++ (Mao et al., 2025). An instruction-based image creation and editing framework with context-aware content filling. ACE++ extends the original ACE architecture with a unified framework for both reference-based generation and local editing, supporting subject-driven object insertion via multi-modal prompts. OmniPaint (Yu et al., 2025). A recent ICCV 2025 method targeting object-oriented editing through disentangled insertion-removal inpainting. OmniPaint decouples the insertion and removal processes, allowing fine-grained control over both the object identity and its interaction with the background. Insert Anything (Song et al., 2025): A dedicated object insertion method based on Diffusion Transformers (DiT). It leverages attention control strategies to blend the reference object into the specific background context, emphasizing identity preservation during the in-context editing process. OmniControl (Tan et al., 2025): A universal control mechanism for DiTs. It introduces effective spatial control injection (e.g., bounding boxes or keypoints) into pre-trained DiT models using minimal additional parameters, enabling precise layout guidance for subject-driven generation.

---

**Algorithm 2:** Automated Multi-Object Dataset Synthesis Pipeline

---

    **Input:** Raw Background $I_{bg}^{raw}$, Target Subject $I_{sub}$, Count $N$
    **Output:** Final Composite Image $I_{final}$
    `// Phase I: Batch Layout Planning`
1  Initialize layout queue $\mathcal{Q} \leftarrow \varnothing$
2  Compute distance field $\mathcal{D} \leftarrow \text{EDT}(M_{valid})$ based on $I_{bg}^{raw}$
3  Initialize occupancy mask $M_{occ} \leftarrow \mathbf{0}$                              `// Track used space`
4  **for** $k \leftarrow 1$ **to** $N$ **do**
      `// Find best spot in unoccupied regions`
5      $p_k \leftarrow \arg\max(\mathcal{D} \cdot (1 - M_{occ}))$
6      **if** $\mathcal{D}(p_k) <$ *threshold* **or** *IsOccupied*$(p_k)$ **then**
7        **break**                          `// Stop if no valid space remains`
8      **end**
      `// Align orientation with topology`
9      $\theta_k \leftarrow \text{ComputeGradientAlignment}(\nabla\mathcal{D}, p_k)$
      `// Update occupancy to prevent overlap`
10     $M_{occ} \leftarrow \text{UpdateOccupancy}(M_{occ}, p_k, \theta_k, \text{size})$
11     Enqueue $\mathbf{p}_k = (p_k, \theta_k)$ into $\mathcal{Q}$
12 **end**
    `// Phase II: Recursive Generative Injection`
13 Initialize state $I_{curr} \leftarrow I_{bg}^{raw}$
14 **while** $\mathcal{Q}$ *is not empty* **do**
15     Pop $\mathbf{p}_k = (p_k, \theta_k)$ from $\mathcal{Q}$
      `// Step 2.1: Context & Subject Prep`
16     $M_k \leftarrow \text{CreateMask}(\mathbf{p}_k)$;    $I_{ctx} \leftarrow I_{curr} \odot (1 - M_k)$
17     $I_{sub}^{rot} \leftarrow \text{Rotate}(I_{sub}, \theta_k)$
      `// Step 2.2: Generation`
18     $x_{gen} \leftarrow \text{Generator}(I_{ctx}, I_{sub}^{rot})$
19     $x_{refined} \leftarrow \text{NATA}(x_{gen}, I_{curr})$                  `// Texture refinement`
      `// Step 2.3: State Update (Loop Closure)`
20     $I_{curr} \leftarrow I_{curr} \odot (1 - M_k) + x_{refined} \odot M_k$
21 **end**
22 **return** $I_{final} \leftarrow I_{curr}$

---

## A.4. Qualitative Visualization and Analysis

In this section, we provide additional qualitative results to visually analyze the inner workings and generation quality of our proposed `PDA` framework.

**Attention Mechanism Visualization.** To understand how `PDA` preserves object identity and integrates it into the context, we visualise the attention maps from the DiT blocks in Figure 6. The self-attention map highlights the model's focus on the structural details of the reference subject. Crucially, the cross-attention map demonstrates how the model attends to the subject's topology while generating the final image within the background context, verifying the effectiveness of our physics-aware guidance.

**Gallery of Generated Samples.** Figure 7 presents a gallery of representative generation results produced by `PDA`. These samples showcase the model's ability to achieve high geometric fidelity, photorealistic lighting harmonization (via the ASA module), and seamless texture assimilation (via the NATA module).

## A.5. Computational Cost Analysis

This appendix provides a detailed cost analysis of PDA, addressing its practical applicability for large-scale data augmentation.

**Per-Stage Overhead** We measure the wall-clock time and peak GPU memory contributed by each PDA module on a single NVIDIA A100 (80 GiB), generating $512 \times 512$ samples one at a time. Results are reported in Table 5.

NATA introduces only a marginal overhead ($+0.4$ s, $+0.40$ GiB) because its Gram-matrix computation is performed entirely

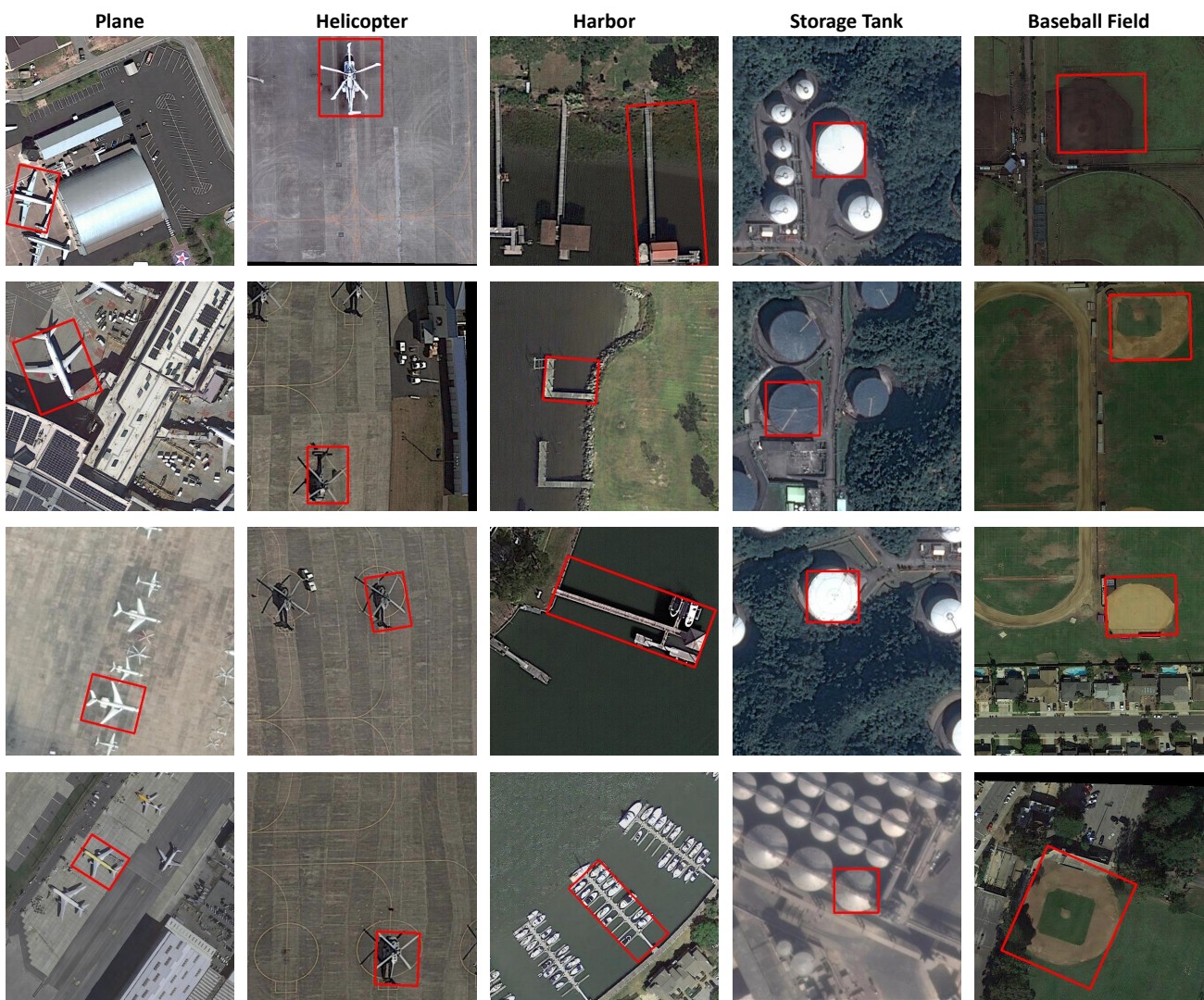

*Figure 7.* Some generated images. The area inside the red box is the one we inserted and generated, the positions, backgrounds and subject are provided by iSAID (Waqas Zamir et al., 2019)

.

in the latent space rather than the pixel or feature space, avoiding any backbone re-evaluation. ASA introduces a larger but still moderate overhead ($+3.3\,\mathrm{s}$, $+2.38\,\mathrm{GiB}$), primarily due to the dual-LoRA branch and the solar context projection. The Planner runs once per background ($0.74\,\mathrm{s}$) and its output can be cached, so for batch synthesis its amortized cost is negligible.

**End-to-End Comparison**  We compare end-to-end inference cost against the strongest available baselines under identical hardware and resolution settings (Table 8).

PDA is faster than Insert Anything ($18.2\,\mathrm{s}$ vs. $21.7\,\mathrm{s}$) while reducing FID by $15.7\%$. Compared to the OminiControl baseline (from which PDA is fine-tuned), the additional cost is $3.7\,\mathrm{s}$ per sample in exchange for a substantial gain in physical fidelity.

**Training Cost**  PDA fine-tunes the OminiControl backbone with LoRA branches for 20,000 steps on a single A100, requiring approximately 39.7 hours of training time. This is comparable to standard OminiControl fine-tuning (34.7 hours under the same setup), representing only a $1.14\times$ overhead. For offline data augmentation pipelines, this one-time training cost is amortized across all subsequent downstream training runs.

*Table 8.* End-to-end inference cost vs. baselines. PDA achieves the lowest FID while remaining faster than Insert Anything and within 1.26× the cost of OminiControl.

| Method | Time (s) ↓ | Mem. (GiB) ↓ | FID ↓ |
|---|---|---|---|
| UniCombine | **6.3** | 34.51 | 22.06 |
| OminiControl | 14.5 | **33.51** | 12.05 |
| Insert Anything | 21.7 | 38.40 | 11.54 |
| PDA (Ours) | 18.2 | 36.30 | **9.732** |

The best results are in **bold** and the second-best are underlined.

### A.6. Cross-Modality and Cross-Sensor Experiments

This appendix provides full results for the cross-modality and cross-sensor evaluation summarized in Section 5.4 of the main paper. We evaluate PDA on three additional benchmarks spanning a different sensor modality (SAR), a wide range of spatial resolutions (sub-meter to 15 m), and a fundamentally different target type (amorphous forest regions vs. rigid vehicles).

#### A.6.1. HRSID: SAR GENERATIVE QUALITY

The High-Resolution SAR Images Dataset (HRSID) (Wei et al., 2020) is a Sentinel-1 SAR ship detection benchmark, representing a fundamentally different sensor modality from our optical training data. Images are tiled into $256 \times 256$ patches, with 3,600 patches used for fine-tuning and 500 patches reserved for evaluation. PDA is fine-tuned without any architectural change, swapping only the upstream segmenter for one trained on SAR data.

As shown in Table 9, PDA improves over the OminiControl baseline across all generative metrics on SAR data, despite the ASA module being originally motivated by optical illumination. We attribute this to the modality-agnostic nature of $\mathbf{c}_{\text{solar}}$: in SAR imagery, the same context vector captures backscattering statistics and local intensity patterns rather than solar geometry, but its function as a sensor-specific prior remains intact. Similarly, NATA's Gram-matrix matching captures SAR speckle statistics in the same way it captures optical micro-textures, since both follow stationary distributions in the latent space.

*Table 9.* Quantitative comparison on HRSID for SAR object insertion. We report both whole-image and insertion-region metrics to evaluate global generation fidelity and local insertion quality, respectively.

| Method Variant | Whole Image | | | | Insertion Region | | |
|---|---|---|---|---|---|---|---|
| | PSNR ↑ | SSIM ↑ | LPIPS ↓ | FID ↓ | PSNR ↑ | SSIM ↑ | LPIPS ↓ |
| Baseline (OminiControl) | 24.45 | 0.7298 | 0.0796 | 21.86 | 14.93 | 0.6739 | 0.0486 |
| + NATA | 24.64 | 0.7292 | 0.0776 | 22.09 | 15.13 | 0.6815 | 0.0481 |
| + ASA | 25.12 | 0.7641 | 0.0684 | **19.78** | 15.28 | 0.6899 | 0.0482 |
| PDA (Ours) | **25.16** | **0.7643** | **0.0682** | 20.03 | **15.36** | **0.6942** | **0.0477** |

The best results are highlighted in **bold**, and the second-best are underlined.

#### A.6.2. SSDD: ZERO-SHOT SAR SHIP DETECTION

To further verify cross-dataset generalization within the SAR modality, we apply PDA *zero-shot* to the SSDD (Zhang et al., 2021) ship detection benchmark (resolutions ranging from 1 m to 15 m, 50 training / 100 validation / 250 test images). PDA is trained on HRSID and used directly to augment the SSDD training set without any further fine-tuning.

As shown in Table 10, PDA achieves the best mAP50 across all four detectors on SSDD without fine-tuning, demonstrating zero-shot cross-dataset generalization on SAR at the task level. The gains are consistent with our optical results in Table 3, confirming that generation-quality improvements translate into real detection gains across modalities and resolutions.

*Table 10.* Downstream ship detection performance (mAP50 in %) on SSDD under the zero-shot SAR augmentation setting. PDA is trained on HRSID and directly applied to SSDD without additional fine-tuning. CP denotes Copy-Paste, CM denotes CutMix, and OC denotes OminiControl. Δ Over Real denotes the improvement of PDA over the real-data baseline.

| Detector | Real | +CP | +CM | +OC | PDA (Ours) | Δ Over Real |
|---|---|---|---|---|---|---|
| Rotated R-CNN | 77.91 | 77.01 | 66.01 | 78.84 | **78.86** | +0.95 |
| Oriented R-CNN | 80.47 | 80.00 | 80.00 | 87.97 | **88.60** | +8.13 |
| S2ANet | 77.36 | 65.44 | 75.29 | 77.86 | **78.42** | +1.06 |
| YOLO26 | 79.62 | 78.93 | 79.30 | 81.57 | **84.28** | +4.66 |
| Avg. | 78.84 | 75.35 | 75.15 | 81.56 | **82.54** | +3.70 |

The best results are highlighted in **bold**, and the second-best are underlined.

### A.6.3. WHU-OPT-SAR: AMORPHOUS TARGET INSERTION

To examine PDA's behavior on objects with fuzzy rather than rigid boundaries, we conduct a forest insertion experiment on the WHU-OPT-SAR (Li et al., 2022) dataset, which has a 5 m spatial resolution. Forest regions are randomly erased using the segmentation labels, and PDA is trained to reconstruct them from reference patches, using 8,616 patches for training and 500 for testing. Results are reported in Table 11, and Figure 8 shows representative examples of forest insertion on WHU-OPT-SAR. This experiment makes two points. First, PDA extends to amorphous targets with fuzzy boundaries, which relaxes the rigid-object assumption behind the original formulation. Second, it works at moderate resolution (5 m), well beyond the very-high-resolution regime of the optical training data. One limitation remains: amorphous targets such as forests would benefit from a soft blending mask rather than a binary insertion mask $M$, and from adapting NATA to handle gradient boundary regions. We leave these refinements to future work.

### A.6.4. SUMMARY OF GENERALIZATION CAPABILITIES

Combining the experiments in this appendix with the optical results in the main paper, we have validated PDA on:

- **Multiple modalities**: optical RGB and SAR.

- **Diverse target types**: rigid objects (aircraft, ships, vehicles) and amorphous objects (forest).

- **Wide resolution range**: from sub-meter (DOTA) to 15 m (SSDD/WHU-OPT-SAR).

- **Task-level validation**: both generation-quality metrics and downstream detection performance.

These results suggest that PDA's modular, physically grounded factorization captures structure shared across remote sensing modalities, rather than structure specific to optical aerial imagery.

### A.7. Failure Cases Analysis

We conduct a systematic failure-mode analysis on the MAR20 zero-shot evaluation to characterize the practical reliability of PDA's implicit solar context modeling. We manually inspect 300 generated images and classify shadow orientation as *correct* (consistent with the scene's solar azimuth) or *incorrect/missing*. We find that 93% of cases exhibit correct shadow orientation, while 7% show identifiable failures. We characterize the two dominant failure modes below and Figure 9 visualizes the two failure modes.

**Failure Mode 1: Bare Backgrounds with No Shadow-Casting Cues.** The implicit solar context vector $\mathbf{c}_{\mathrm{solar}}$ is extracted by aggregating high-level features from the masked background latent $z_{\mathrm{bg}}$. When the background contains no objects casting shadows—e.g., open runways, empty tarmacs, or featureless ground—there is insufficient visual evidence for the model to infer the scene's solar azimuth. In these cases, $\mathbf{c}_{\mathrm{solar}}$ collapses to an under-specified prior, leading to either omitted shadows or shadows pointing in arbitrary directions.

**Failure Mode 2: Extreme Background Conditions.** In images with extreme conditions (e.g., heavy cloud cover, snow, deep shadows over the entire scene), the high-level features that $\mathbf{c}_{\mathrm{solar}}$ relies on become unreliable. Since generalization

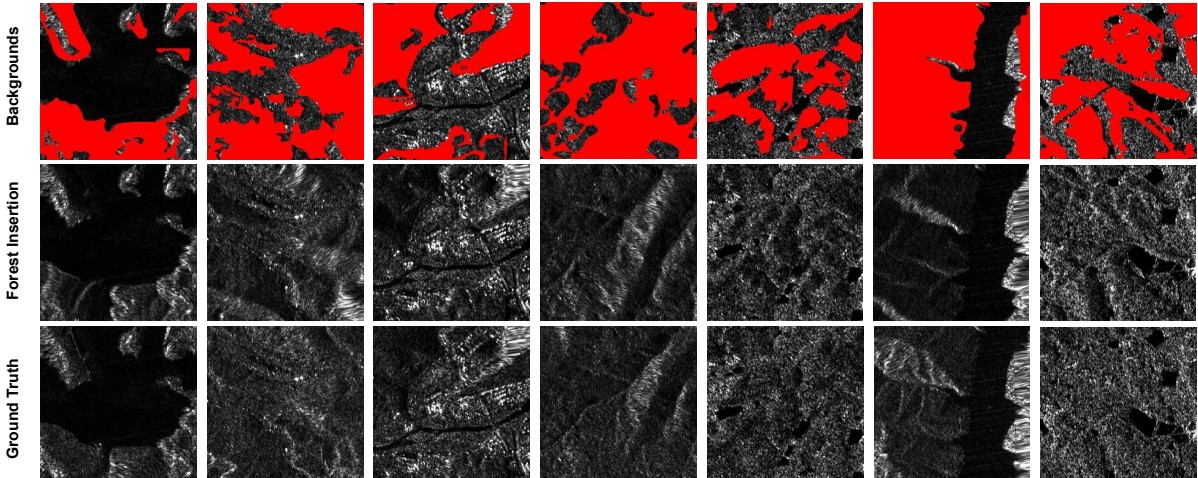

*Figure 8.* Representative examples of forest insertion on WHU-OPT-SAR. The first row shows the input backgrounds, where the red regions indicate the target areas to be inserted or reconstructed. The second row shows the corresponding forest insertion results generated by PDA, and the third row shows the ground-truth images. These examples illustrate that PDA can handle amorphous targets with fuzzy boundaries in the SAR domain.

*Table 11.* Quantitative comparison on WHU-OPT-SAR for amorphous target insertion. We report both whole-image and insertion-region metrics to evaluate global reconstruction fidelity and local insertion quality, respectively.

| Method Variant | Whole Image | | | | Insertion Region | | |
|---|---|---|---|---|---|---|---|
| | PSNR ↑ | SSIM ↑ | LPIPS ↓ | FID ↓ | PSNR ↑ | SSIM ↑ | LPIPS ↓ |
| Baseline (OminiControl) | 18.72 | 0.5958 | 0.1644 | 32.47 | 18.15 | 0.5830 | 0.1901 |
| + NATA | 18.75 | 0.5963 | 0.1642 | 32.11 | 18.22 | 0.5841 | 0.1899 |
| + ASA | 18.83 | 0.5998 | 0.1616 | 31.15 | 18.28 | 0.5885 | 0.1829 |
| PDA (Ours) | **18.87** | **0.6002** | **0.1615** | **30.88** | **18.37** | **0.5892** | **0.1828** |

The best results are highlighted in **bold**, and the second-best are underlined.

of the implicit context is bounded by the diversity of the training distribution, such out-of-distribution backgrounds yield degraded shadow synthesis even when shadow-casting cues are present.

**Practical Implications.** Despite these failure modes, PDA still delivers consistent gains across all four downstream detectors (Table 3). This is because Copy-Paste, the de facto baseline, performs no illumination adjustment whatsoever, leaving shadow consistency entirely to chance. PDA's 93% physically consistent rate is therefore a clear positive augmentation signal even in the presence of the failure cases described above. We see two natural directions for further reducing this 7% rate: (i) learning an explicit conditional distribution over solar azimuth from rich training scenes and applying it as a prior when $c_{solar}$ is under-specified; and (ii) adding self-consistency regularization that penalizes shadow disagreement among multiple synthesized samples on the same background. Both are out of scope for the present work.

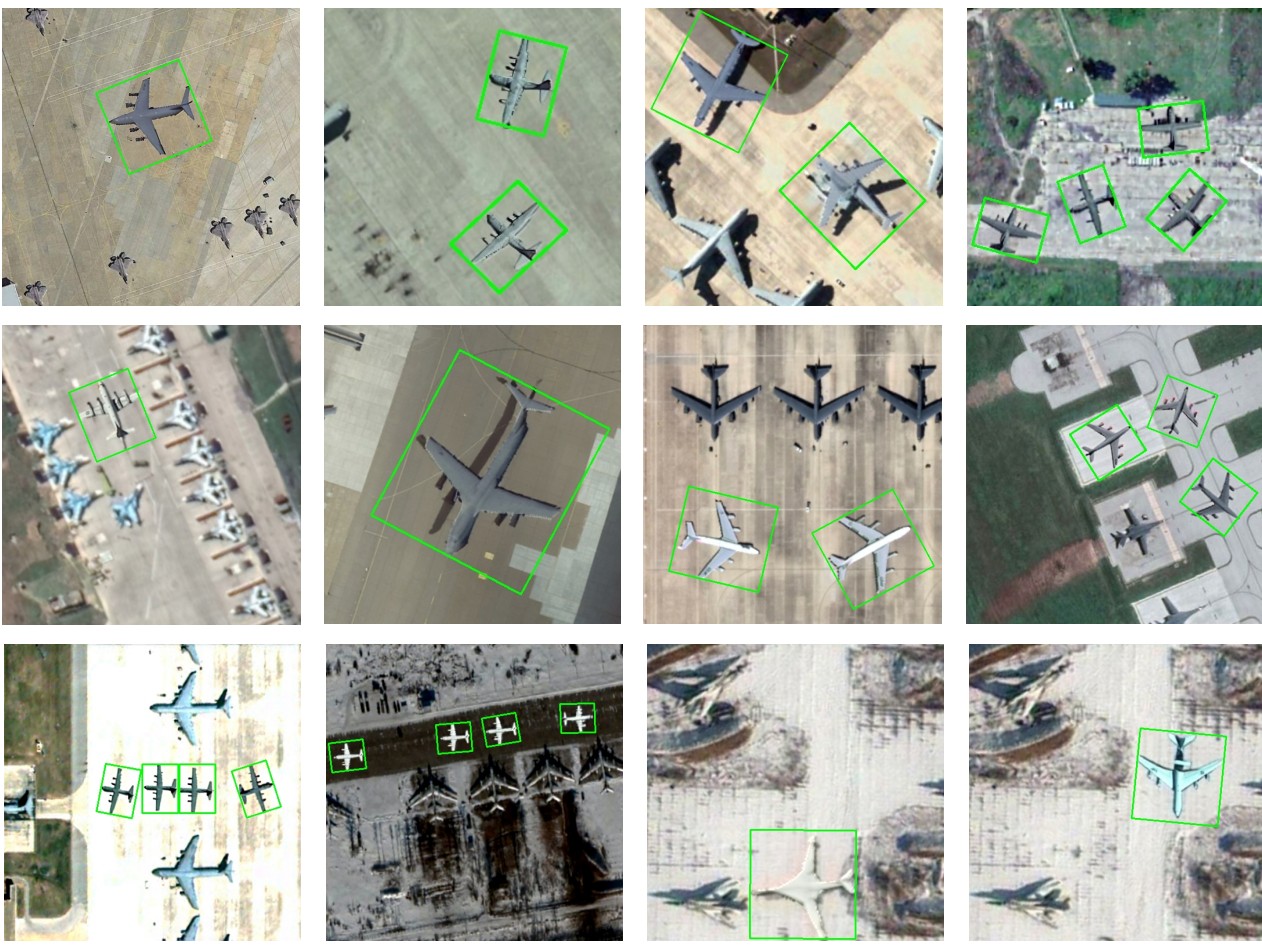

*Figure 9.* Visualization of PDA failure cases on MAR20 zero-shot evaluation. **Top:** bare-background failures. **Bottom:** extreme-condition failures. See Appendix A.7 for full analysis.

