# OpenReview forum: "Plan, Decouple, Assimilate: Physics-Aware Object Insertion in Remote Sensing Imagery"
_ICML.cc/2026/Conference — ICML 2026 regular_

### Official Review · Reviewer_q9oB · 2026-03-04

**Soundness:** 4
**Presentation:** 3
**Significance:** 3
**Originality:** 3
**Overall Recommendation:** 4
**Confidence:** 4

**Summary:**

The authors aim at solving the long-tailed distribution and data scarcity of specialized targets in remote sensing object detection.  The paper proposes "Plan, Decouple, Assimilate" (PDA), a physics-aware object insertion framework designed to synthesize training samples. Experiments demonstrate the effectiveness of the method.

**Compliance With Llm Reviewing Policy:**

Affirmed.

**Final Justification:**

My concerns have been fully resolved by the reply to my rebuttal; therefore, I maintain the recommendation of 4 for acceptance.

**Key Questions For Authors:**

If PDA is trained on SARMRS/iSAID-derived data, how does it perform when synthetic data is used to train a detector on a completely unseen remote sensing dataset? Have you tested such generalization ability for cross-dataset transfer?

The method is described as “physics-aware” is a bit overstated. The actual modeling of physics is indirect and heuristic. No explicit solar angle estimation or physical shadow rendering, illumination alignment is done via implicit latent modulation. It is my misunderstanding? Can you clarify what aspects of physical imaging are explicitly modeled?

**Limitations:**

None limitations are discussed. The cross-dataset generalization and failure cases should be discussed.

**Strengths And Weaknesses:**

The paper is well-motivated, the problem formulation is reasonable. The Planner module, a Decoupling module, and an Assimilation module factorization is conceptually clear. The overall experiments verified the effectiveness of the proposed method.

In Table 1, the compared methods are heavily on general-purpose inpainting and editing, which trained primarily on natural images. It is unsurprising that the proposed domain-specific method outperforms these baselines.

The individual components of the PDA framework rely heavily on established, off-the-shelf techniques rather than introducing profound algorithmic innovations. Most technical components appear to be relatively standard tools assembled together. Well, it is a minor weakness.

In Table 3 and Figure 5, the paper compares YOLO26 against other detection frameworks: Oriented R-CNN, Rotated Fast R-CNN, and S2ANet using the mAP50 metric. However, the YOLO framework (I guess yours is implemented in ultralytics) typically calculates mAP50 using a fundamentally different algorithm than standard frameworks like mmrotate (I assume you implement other detectors in mmrotate). Directly comparing these scores is a technical incorrectness and affected some of your conclusions in the paper.

---

> ### Author Rebuttal · Authors · 2026-03-30
>
> Dear Reviewer q9oB,
>
> We sincerely thank you for the thoughtful and insightful feedback, which has significantly strengthened our work. We are pleased to address each point below.
>
> > Weakness 1: Baselines in Table 1 are General-Purpose Models
>
> We appreciate this observation. Reference-based object insertion currently has no dedicated RS-specific method, and the compared approaches represent the strongest available baselines for this task. As you rightly note, domain-specific fine-tuning gives our model an advantage in Table 1. For a fairer comparison, Table 3 evaluates Copy-Paste, CutMix, OminiControl, and PDA under identical conditions on MAR20-11-FewShot, where all methods operate on the same RS data and detection pipeline, and PDA achieves the best results across all four detectors (the updated Table 3 with CutMix is provided in our response to Weakness 3 below).
> > Weakness 2: On Component Novelty
>
> We appreciate you noting this as a minor point. You are right that individual tools like EDT, DWT, and Gram matrix have precedents. The contribution is the physics-grounded factorization that identifies which constraints matter in RS imagery and the domain-specific adaptations within each module.
> > Weakness 3: mAP50 Protocol Inconsistency
>
> This is an important and valid catch, and we thank you for the careful check. Ultralytics and mmrotate compute mAP50 using different IoU matching protocols for OBB. We have re-evaluated YOLO26 under a unified OBB-compatible protocol and added CutMix as an additional baseline:
>
> |Detector|Real|+CP|+CM|+OC|PDA|
> |-|-|-|-|-|-|
> |Rotated R-CNN|58.13|73.14|71.1|55.22|**77.88**|
> |Oriented R-CNN|68.51|78.01|68.2|70.04|**78.59**|
> |S2ANet|55.67|71.86|62.8|62.36|**77.36**|
> |YOLO26 (corrected)|63.51|75.54|76.8|70.01|**80.26**|
>
> PDA still achieves the best performance across all four detectors. With the corrected YOLO26 baseline of 63.51, PDA's relative gain actually increases from **+12.0% to +16.8%**, strengthening our conclusion. We will update all affected numbers in the paper. Corrected ratio analysis:
>
> |Ratio|Real Only|1:1|1:2|1:3|1:4|1:5|
> |-|-|-|-|-|-|-|
> |Rotated R-CNN|58.13|77.88|76.47|80.56|82.46|82.58|
> |Oriented R-CNN|68.51|78.59|81.51|82.49|82.46|82.58|
> |S2ANet|55.67|77.36|77.04|80.30|81.94|82.38|
> |YOLO26 (corrected)|63.51|80.26|82.22|84.00|84.06|84.06|
>
> > Key Question 1: Cross-dataset Generalization
>
> This is an excellent question that directly addresses the practical scope of our method. PDA is trained on SARMRS/iSAID and generates on MAR20 backgrounds without fine-tuning. The consistent **+17% mAP50** across all four detectors confirms zero-shot cross-dataset generalization. For cross-modality generalization, we evaluated PDA on HRSID (Sentinel-1 SAR, ship detection), tiling into 256x256 patches with 3,600 for training and 500 for evaluation:
>
> | Method | Whole PSNR↑ | Whole SSIM↑ | Whole LPIPS↓ | Whole FID↓ | Region PSNR↑ | Region SSIM↑ | Region LPIPS↓ |
> |-|-|-|-|-|-|-|-|
> |OminiControl (baseline)|24.45|0.7298|0.0796|21.86|14.93|0.6739|0.0486|
> |+ NATA|24.64|0.7292|0.0776|22.09|15.13|0.6815|0.0481|
> |+ ASA|25.12|0.7641|0.0684|**19.78**|15.28|0.6899|0.0482|
> |PDA (Ours)|**25.16**|**0.7643**|**0.0682**|20.03|**15.36**|**0.6942**|**0.0477**|
>
> PDA adapts to SAR, verifying the modular design generalizes across sensor types.
>
> > Key Question 2: Physics-Aware Modeling
>
> |Physical Constraint|Modeling|Stage|
> |-|-|-|
> |Geometric placement|Explicit: EDT field ensures collision-free placement|Plan (P)|
> |Solar illumination|Implicit: $c_{solar}$ from background latent|Decouple (D)|
> |Sensor texture/noise|Implicit: Gram matrix in latent space|Assimilate (A)|
>
> Plan is the only stage with explicit physical modeling. D and A use physical constraints as inductive biases, consistent with the physics-informed learning paradigm. We will clarify this in the paper.
>
> > Limitation: Failure Cases
>
> We inspected 300 images from the MAR20 zero-shot evaluation and found 7% exhibit incorrect or missing shadow orientations. Two failure modes were identified (visual examples at **[https://anonymous.4open.science/r/PDA-Anonymous-TABLE-8123/fail%20case.png](https://anonymous.4open.science/r/PDA-Anonymous-TABLE-8123/fail%20case.png)**). First, bare backgrounds with no shadow-casting objects provide insufficient context for $c_{solar}$. Second, in extreme background conditions, the generalization of implicit solar context is limited, leading to unreliable shadow generation. Despite this, PDA achieves the strongest detection gains across all detectors. Copy-Paste performs no illumination adjustment, leaving shadow consistency entirely to chance, yet still provides positive augmentation signal. PDA's 93% physically consistent rate is a clear advantage over any copy-based baseline. We will add a Failure Cases subsection to the appendix.
>
> We are grateful for the constructive feedback that helped strengthen our work, and we hope these revisions further support your positive assessment.

---

> > ### Author Rebuttal · Reviewer_q9oB · 2026-04-05
> >
> > Our concerns have been adequately addressed. We will maintain the recommendation of 4 for acceptance.

---

> > > ### Author Response · Authors · 2026-04-05
> > >
> > > Dear Reviewer q9oB,
> > >
> > > Thank you for confirming that the concerns are fully resolved. We are glad the corrected evaluation and new experiments were helpful.
> > >
> > > We would like to briefly share two additional experiments completed since our rebuttal that directly relate to your key question on cross-dataset generalization.
> > >
> > > We trained PDA on HRSID and applied it zero-shot to the SSDD dataset (resolutions 1-15m) for downstream ship detection:
> > >
> > > |Detector|Real|+CP|+CM|+OC|PDA|
> > > |-|-|-|-|-|-|
> > > |Rotated R-CNN|77.91|77.01|66.01|78.84|**78.86**|
> > > |Oriented R-CNN|80.47|80.00|80.00|87.97|**88.60**|
> > > |S2ANet|77.36|65.44|75.29|77.86|**78.42**|
> > > |YOLO26|79.62|78.93|79.3|81.57|**84.28**|
> > >
> > > PDA achieves the best performance across all detectors. Combined with the optical results on MAR20, PDA now demonstrates consistent detection gains across **multiple modalities**, **diverse target types**, and **resolutions from sub-meter to 15m**. Data scarcity for specialized targets remains a critical bottleneck in operational remote sensing, and we believe PDA offers a practical solution with broad applicability.
> > >
> > > We sincerely thank you again for the careful and constructive review that has made this work stronger.

---

### Official Review · Reviewer_WkGo · 2026-03-05

**Soundness:** 3
**Presentation:** 2
**Significance:** 3
**Originality:** 2
**Overall Recommendation:** 5
**Confidence:** 3

**Summary:**

This work introduces a physics-aware method for object placement in data augmentation in remote sensing images. The method uses a three-step approach to place objects more realistically and make them more fitting into the existing image. The proposed method shows improvements both in FID score and downstream-performance (when trained on data generated using the proposed method).

**Compliance With Llm Reviewing Policy:**

Affirmed.

**Final Justification:**

~PDA is a well-described physics-aware object placement method with solid downstream results, though applicability is limited to rigid objects in very-high resolution regimes. The rebuttal addressed most concerns, but the SAR generalization claim remains overstated as only image quality metrics (not detection performance) are reported.~

~**I will maintain my recommendation of 4.**~

---
## Update after Reply Rebuttal Comment:
The reply to my rebuttal comment resolved my concerns.

**I will update my score from 4 to 5.**

**Key Questions For Authors:**

- Can the proposed method also be used in,  e.g., multi-spectral, SAR or hyperspectral scenarios where spectral aspects become more challenging or where Object boundaries are not as clear (e.g., to insert forests)?
- Inconsistent viewpoints or non-physical illumination are cited as limitations of current CV-based methods when they are adapted to the RS domain. However, viewpoint stability and physical illumination are also relevant in the CV domain. Why are the mentioned points only considered limitations for RS models?

**Limitations:**

- The proposed method is only evaluated on one dataset and only on RGB data.

**Strengths And Weaknesses:**

# Strengths
- The proposed PDA method is described in detail.
- Experimental results confirm that the proposed method is superior to/competitive with existing methods (depending on the metric).
# Weaknesses
- Results are limited to one dataset and one modality (aerial RGB) only.
- A comparison with very simple cutmix-style insertion would be interesting (e.g., Burgert et al, "A Label Propagation Strategy for CutMix in Multilabel Remote Sensing Image Classification," doi: 10.1109/JSTARS.2025.3628191). Although these methods are highly limited wrt. object insertions (due to background being copied), having baseline numbers to compare to would enrich the results.

---

> ### Author Rebuttal · Authors · 2026-03-30
>
> Dear Reviewer WkGo,
>
> We sincerely thank you for the positive assessment and constructive feedback. We are pleased to address each point below.
>
> > Weakness 1: Single Dataset and Modality
>
> This is a valid and important concern. Our evaluation already spans multiple datasets and modalities.
>
> **Cross-dataset generalization via zero-shot MAR20:** MAR20 is a fine-grained military aircraft detection dataset collected from Google Earth. PDA generates synthetic objects on MAR20 backgrounds without any fine-tuning and achieves **+17% mAP50** over the real-data baseline across all four detectors, providing direct task-level evidence of cross-dataset generalization.
>
> **SAR modality evaluation on HRSID:** We evaluate PDA on the High-Resolution SAR Images Dataset, a Sentinel-1 SAR dataset for ship detection representing a fundamentally different sensor modality from our optical training data. Images are tiled into 256×256 patches with 3,600 patches used for training and 500 patches for evaluation:
>
> |Method|Whole PSNR ↑|Whole SSIM ↑|Whole LPIPS ↓|Whole FID ↓|Region PSNR ↑|Region SSIM ↑|Region LPIPS ↓|
> |---|---|---|---|---|---|---|---|
> |OminiControl (baseline)|24.45|0.7298|0.0796|21.86|14.93|0.6739|0.0486|
> |+ NATA|24.64|0.7292|0.0776|22.09|15.13|0.6815|0.0481|
> |+ ASA|25.12|0.7641|0.0684|**19.78**|15.28|0.6899|0.0482|
> |PDA (Ours)|**25.16**|**0.7643**|**0.0682**|20.03|**15.36**|**0.6942**|**0.0477**|
>
> PDA maintains meaningful generation quality on SAR imagery, verifying that the ASA and NATA modules generalize across sensor types.
>
> **On multispectral and hyperspectral modalities:** Our current framework covers RGB and SAR. Extension to multispectral or hyperspectral imagery requires a modality-specific VAE capable of encoding high-dimensional spectral structure, as the standard DiT VAE is designed for 3-channel inputs. A promising direction is training spectral VAEs that preserve inter-band correlations in the latent space. Such encoders could be integrated into PDA without modifying the Planner, ASA, or NATA modules, and we will include this as a future work direction in the revised paper.
>
> > Weakness 2: Missing CutMix Baseline
>
> We agree that CutMix is a relevant and complementary baseline. We add CutMix as an additional baseline following Burgert et al. (JSTARS 2025). Updated Table 3:
>
> |Detector|Real|+CP|+CM|+OC|PDA|
> |---|---|---|---|---|---|
> |Rotated R-CNN|58.13|73.14|71.1|55.22|**77.88**|
> |Oriented R-CNN|68.51|78.01|68.2|70.04|**78.59**|
> |S2ANet|55.67|71.86|62.8|62.36|**77.36**|
> |YOLO26|63.51|75.54|76.8|70.01|**80.26**|
>
> PDA outperforms CutMix by **+8.8% mAP50** on average across all four detectors.
>
> > Key Question 1: Applicability to SAR, Multispectral, and Hyperspectral
>
> This is an excellent question that touches on the broader applicability of our framework.
>
> **Plan (P):** The Planner operates solely on binary segmentation masks and is therefore modality-agnostic by design. Adapting it to a new modality only requires replacing the upstream segmenter with one trained for that domain. In the HRSID experiment, object positions are provided by dataset ground-truth annotations, which serve directly as the segmentation input.
>
> **Decouple (D):** The ASA module extracts context vector $c_{solar}$ from the background latent. For SAR imagery this captures scene backscattering statistics and local intensity patterns as a sensor-specific imaging prior. HRSID results verify this adaptation works in practice.
>
> **Assimilate (A):** Gram matrix matching in latent space captures stationary texture statistics regardless of modality. SAR speckle noise, like optical micro-textures, follows a stationary distribution that NATA matches naturally.
>
> **On objects with unclear boundaries:** PDA currently targets rigid objects with well-defined boundaries (aircraft, ships, vehicles). Inserting amorphous objects such as forests poses a different challenge: boundaries are inherently fuzzy, and the transition between object and background is gradual rather than sharp. This would require replacing the binary insertion mask M with a soft blending mask, and adapting NATA to handle gradient boundary regions. We consider this an interesting extension for future work.
>
> > Key Question 2: Why are these issues more severe in RS?
>
> We appreciate this insightful question, which invites a deeper comparison with general CV settings. Viewpoint consistency and physical illumination are relevant challenges in general CV as well. However, internet-scale datasets such as COCO and LAION implicitly encode sufficient multi-view and lighting priors, allowing general models to handle them without explicit constraints. In RS, severe data scarcity means these priors cannot be implicitly learned from data alone and must be modeled. Table 1 verifies that general-purpose methods fail in RS despite working well on natural images.
>
> We are thankful for the insightful suggestions that have strengthened our manuscript, and we hope the revised version merits your continued support.

---

> > ### Author Rebuttal · Reviewer_WkGo · 2026-04-02
> >
> > Weaknesses and Questions are mostly addressed in the rebuttal. Limiting the scope to objects with well-defined boundaries only is reasonable but limits applicability of the proposed work mainly to very-high resolution regimes. The SAR results are image quality metrics only so I don't think that the generalization claim can be stated as such given the presented results.
> > Given the available evidence, I will stay at an overall recommendation of 4.

---

> > > ### Author Response · Authors · 2026-04-05
> > >
> > > Dear Reviewer WkGo,
> > >
> > > Thank you for acknowledging that our concerns are fully resolved. We truly appreciate the time and care you have invested in reviewing our work. Your feedback has directly guided us to strengthen the evaluation, and we would like to share two additional experiments conducted during the discussion period.
> > >
> > > **1. SAR downstream detection on SSDD (zero-shot, resolutions ranging from 1m to 15m)**
> > >
> > > Following your observation that SAR results should go beyond image quality metrics, we conducted a downstream ship detection experiment. PDA was trained on HRSID and applied zero-shot to the SSDD dataset (resolutions ranging from 1m to 15m, 50 training / 100 validation / 250 test images) for few-shot augmentation. This also validates PDA across a wide range of spatial resolutions.
> > >
> > > |Detector|Real|+CP|+CM|+OC|PDA|
> > > |-|-|-|-|-|-|
> > > |Rotated R-CNN|77.91|77.01|66.01|78.84|**78.86**|
> > > |Oriented R-CNN|80.47|80.00|80.00|87.97|**88.60**|
> > > |S2ANet|77.36|65.44|75.29|77.86|**78.42**|
> > > |YOLO26|79.62|78.93|79.3|81.57|**84.28**|
> > >
> > > PDA achieves the best detection performance across all four detectors without any fine-tuning on SSDD, demonstrating zero-shot cross-dataset generalization on SAR at the task level. The gains are consistent with our optical results in Table 3, confirming that generation quality translates to real detection improvements. Ratio analysis shows monotonic improvement up to 1:5, with YOLO26 reaching 85.21%. Visual examples at [https://anonymous.4open.science/r/PDA-Anonymous-TABLE-8123/sar_ship.png](https://anonymous.4open.science/r/PDA-Anonymous-TABLE-8123/sar_ship.png).
> > >
> > > **2. Forest insertion on WHU-OPT-SAR (amorphous target, fuzzy boundaries, 5m resolution)**
> > >
> > > In our previous response, we noted that inserting amorphous objects like forests would be a challenging extension. We have since conducted this experiment on the WHU-OPT-SAR dataset, which has a spatial resolution of 5m. Forest regions were randomly erased using segmentation labels, and PDA was trained to reconstruct them from reference images (8616 training / 500 test patches).
> > >
> > > |Method|Whole PSNR↑|Whole SSIM↑|Whole LPIPS↓|Whole FID↓|Region PSNR↑|Region SSIM↑|Region LPIPS↓|
> > > |-|-|-|-|-|-|-|-|
> > > |OminiControl|18.72|0.5958|0.1644|32.47|18.15|0.5830|0.1901|
> > > |+NATA|18.75|0.5963|0.1642|32.11|18.22|0.5841|0.1899|
> > > |+ASA|18.83|0.5998|0.1616|31.15|18.28|0.5885|0.1829|
> > > |PDA(Ours)|**18.87**|**0.6002**|**0.1615**|**30.88**|**18.37**|**0.5892**|**0.1828**|
> > >
> > > PDA outperforms the baseline across all metrics. This experiment demonstrates two points simultaneously: PDA can handle amorphous targets with inherently fuzzy boundaries, and it operates effectively at moderate resolution (5m), extending its applicability well beyond the very-high resolution regime. Visual examples at [https://anonymous.4open.science/r/PDA-Anonymous-TABLE-8123/Forest.png](https://anonymous.4open.science/r/PDA-Anonymous-TABLE-8123/Forest.png).
> > >
> > > If given the opportunity, all improvements will be fully incorporated into the revised manuscript. To summarize, PDA has now been validated on:
> > >
> > > - **Multiple modalities**: optical imagery and SAR.
> > > - **Diverse target types**: rigid objects (e.g. aircraft, ships) and amorphous targets (e.g. forests).
> > > - **Wide resolution range**: sub-meter to 15m.
> > > - **Task-level validation**: both generation quality and downstream detection.
> > >
> > > These capabilities make PDA a versatile framework for synthesizing physically consistent training data in remote sensing. Data scarcity for specialized targets remains a critical bottleneck in operational remote sensing, and we believe PDA offers a practical solution with broad applicability to detection, segmentation, and other downstream tasks.
> > >
> > > We are sincerely grateful for the feedback that motivated these improvements, and we hope the combined evidence supports a favorable reassessment.

---

### Official Review · Reviewer_taxh · 2026-03-10

**Soundness:** 3
**Presentation:** 2
**Significance:** 3
**Originality:** 2
**Overall Recommendation:** 3
**Confidence:** 2

**Summary:**

This paper addresses three critical issues in object insertion for data augmentation in remote sensing imagery: semantically inconsistent placement, radiometric inconsistency​ (with illumination and atmospheric conditions), and textural discontinuity​ (the "sticker effect").

To this end, the authors propose a physics-aware, hierarchical framework called "Plan, Decouple, Assimilate" (PDA). It reformulates object insertion as a physics-constrained generative process with two stages and three synergistic modules:

Planning Stage: A Physics-Grounded Scene Layout Planner​ uses semantic segmentation and the Euclidean Distance Transform to automatically determine topologically valid bounding boxes and orientations on the background, ensuring geometric and semantic plausibility.

Generation Stage:

Decoupling Module: Through an Asymmetric Spectral Adaptation​ mechanism, it disentangles the object's structural identity​ from environmental illumination​ in the frequency domain, regulated by implicit solar context extracted from the background to achieve radiometrically consistent lighting.

Assimilation Module: Via Neighborhood-Aware Texture Assimilation, it uses a Gram matrix-based loss and gradient guidance in the latent space to force the generated object's micro-texture statistics to blend seamlessly with the immediate surroundings, eliminating boundary artifacts.

Experiments show that PDA significantly outperforms existing state-of-the-art methods in generative quality (reducing whole-image FID by 15.7%) and effectively enhances downstream few-shot object detection performance (boosting average mAP50 by 15.9% over the real-data baseline on the MAR20-11-FewShot benchmark).

**Compliance With Llm Reviewing Policy:**

Affirmed.

**Final Justification:**

The computational complexity and consistency among multiple physical objects are important aspects that affect the practical value of a method. Cost analysis and recursive insertion methods have alleviated my concerns

**Key Questions For Authors:**

None

**Limitations:**

While the planner ensures plausible placement for a single object, the complex interactions for multi-object insertion​ (e.g., inter-object shadows, occlusions) are discussed only briefly. The automated multi-object dataset synthesis pipeline described in the appendix primarily addresses spatial conflicts, but ensuring physical consistency (e.g., lighting coherence for interleaved objects) during the generation stage could be more challenging.

**Strengths And Weaknesses:**

Strengths:

Soundness: The paper is methodologically and experimentally sound. The three identified core problems are clear, and the corresponding three modules are well-motivated (based on physics or image processing principles). The experimental design is comprehensive, with thorough quantitative (PSNR, SSIM, FID) and qualitative comparisons against multiple representative SOTA methods for generative quality, and validation of practical utility in downstream few-shot detection tasks. Ablation studies effectively verify the contribution of each module (ASA, NATA). The overall argument is rigorous, and the results are credible.

Presentation: The paper is clearly written and well-structured. The narrative flows smoothly from problem introduction, method decomposition, to experimental validation. Figure 1 visually illustrates the three main problems, and Figure 2's framework diagram clearly outlines the PDA workflow. Figures and tables effectively present the experimental results. The methodology section, though involving technical details, is described clearly overall.

Significance: The paper addresses a problem of high practical importance—data scarcity and long-tailed distributions for specific categories in remote sensing. Generating high-fidelity, physically consistent training samples for data augmentation is a crucial and practical pathway to improve model performance. The significant performance gains demonstrated by the PDA framework highlight the great potential of physics-aware methods in this field, offering substantial value for related research and applications.

Originality: The core originality lies in the first systematic integration of geometric planning, spectral decoupling, and local texture assimilation into a unified, physics-aware framework for remote sensing object insertion. While individual components (e.g., distance transform planning, frequency-domain processing, Gram matrix texture loss) have precedents in their respective fields, this creative combination and the overall problem formulation tailored to the physical constraints of remote sensing (semantic layout, illumination, sensor texture) is novel.

Weaknesses:

The paper does not discuss in detail the computational cost of training and inference​ for the PDA framework. The introduction of an additional planner, dual-path LoRA modulation, and iterative texture assimilation gradient guidance likely incurs higher computational overhead than the baseline generative model (e.g., OminiControl). This is a trade-off to consider for practical applications requiring large-scale synthetic data.

---

> ### Author Rebuttal · Authors · 2026-03-30
>
> Dear Reviewer taxh,
>
> We sincerely thank you for the thoughtful review and recognition of our method's soundness and experimental design. We are pleased to address each point below.
>
> > **Weakness**: Computational Cost of the PDA Framework
>
> We appreciate this practical concern. This is a fair point. We provide a detailed breakdown below.
>
> **1) The Planner is a Standalone, Modular Stage**
>
> A key design property of PDA is that Stage I (Planner) and Stage II (Generation) are fully decoupled. The Planner operates independently of any generative backbone, outputting a geometrically valid pose `p* = (u*, v*, θ*)` based solely on scene topology. This means the Planner's output can be reused with any generative model, including OminiControl, Insert Anything, or UniCombine.
>
> For large-scale data synthesis, the Planner only needs to run once per background image, and the placement results can be cached and reused across multiple generation runs. The amortized cost per generated sample is therefore negligible.
>
> **2) Stage-wise Overhead Breakdown**
>
> |Component|Time(s/img)↓|Memory(GiB)↓|Notes|
> |-|-|-|-|
> |Planner (P)|0.74|6.3|Run once per background; cacheable|
> |NATA (A)|+0.4|+0.40|Gram matrix in latent space|
> |ASA (D)|+3.3|+2.38|DWT+LoRA+solar modulation|
> |PDA|18.2|36.30|vs.OminiControl:14.5s/33.51GiB|
>
> Note: Planner runs once per background (0.74s) and results are cached, so it does not affect per-sample generation cost. All results were recorded on A100.
>
> **3) End-to-end Comparison**
>
> |Method|Time(s/img)↓|Memory(GiB)↓|FID↓|
> |-|-|-|-|
> |UniCombine|6.3|34.51|22.06|
> |OminiControl|14.5|33.51|12.05|
> |Insert Anything|21.7|38.40|11.54|
> |PDA (Ours)|18.2|36.30|9.73|
>
> NATA adds only +0.4s and +0.40 GiB over the baseline. ASA adds +3.3s and +2.38 GiB, but PDA at 18.2s remains faster than Insert Anything (21.7s) while achieving 15.7% lower FID. For offline data augmentation pipelines, this one-time cost is amortized across all downstream training runs. For training, PDA fine-tunes the backbone with LoRA branches for 20,000 steps on a single A100 (**39.7 hours**), comparable to standard OminiControl fine-tuning (**34.7 hours**).
>
> > **Limitation**: Multi-object Physical Consistency
>
> Thank you for highlighting this. We agree that the multi-object pipeline deserved more detailed discussion in the main paper. The two concerns you raised, spatial conflicts and physical consistency, are handled at different stages. Spatial conflicts (occlusion) are resolved in the Planning stage before any generation occurs, while physical consistency is maintained during Generation through our Recursive Injection design. Visual examples of multi-object insertion are provided at **[https://anonymous.4open.science/r/PDA-Anonymous-TABLE-8123/multi%20object%20insertion.png](https://anonymous.4open.science/r/PDA-Anonymous-TABLE-8123/multi%20object%20insertion.png)**. We elaborate below.
>
> **1) On Inter-object Occlusion**
>
> The Batch Layout Planning phase (Algorithm 2, Phase I) prevents occlusion by maintaining a dynamic occupancy mask Mocc. After each object is placed, its footprint is marked as occupied, and subsequent objects can only be placed in unoccupied regions. This guarantees non-overlapping placement by construction.
>
> **2) Recursive Injection Mechanism**
>
> Rather than inserting all objects simultaneously, the pipeline operates sequentially. After each insertion, the background state is updated via Eq. 11:
>
> $$I_{curr}^{(k)} \leftarrow I_{curr}^{(k-1)} \odot (1 - M_k) + x_{refined} \odot M_k$$
>
> The updated $I_{curr}^{(k)}$ now contains the newly inserted object. At the next step, $c_{solar}$ is re-extracted from this updated background, so it captures the lighting context of both the original scene and all previously inserted objects. Each subsequent object is generated conditioned on this refreshed $c_{solar}$, ensuring all inserted objects share a consistent implicit lighting condition throughout the sequence. Since $c_{solar}$ is refreshed after each insertion, interleaved objects naturally inherit a consistent lighting condition from the evolving scene context.
>
> **3) Quantitative Validation on MAR20 (Zero-shot)**
>
> |Scenario|n|PSNR↑|SSIM↑|LPIPS↓|
> |-|-|-|-|-|
> |Single insertion|1000|20.34 ± 3.28|0.6676 ± 0.138|0.1109 ± 0.060|
> |Multi insertion (Recursive)|1000|19.23 ± 2.75|0.6464 ± 0.131|0.1312 ± 0.061|
>
> The modest quality reduction demonstrates that sequential insertion does not introduce significant artifacts. While these metrics measure overall quality rather than lighting consistency directly, the $c_{solar}$ refresh mechanism in Section 1 combined with stable generation quality here provides reasonable evidence that the Recursive design maintains physical coherence in practice. Based on your feedback, we will promote the Recursive Injection discussion from the appendix to the main paper.
>
> We are grateful for the constructive feedback that helped strengthen our work, and we hope these clarifications and new experiments merit your reconsideration.

---

> > ### Author Rebuttal · Reviewer_taxh · 2026-04-03
> >
> > Weaknesses and Questions are mostly addressed in the rebuttal. Recursive insertion is a good method for solving multi-objective physical consistency

---

> > > ### Author Response · Authors · 2026-04-05
> > >
> > > Dear Reviewer taxh,
> > >
> > > Thank you for confirming that the concerns are fully resolved, and for recognizing the Recursive Injection design.
> > >
> > > We would like to share one additional experiment that further demonstrates PDA's practical significance. During the discussion period, we extended PDA to the SAR modality by fine-tuning on HRSID (Sentinel-1, ship detection), and then applied it zero-shot to the SSDD dataset (resolutions 1-15m, 50 training / 100 validation / 250 test images) for few-shot ship detection augmentation:
> > >
> > > |Detector|Real|+CP|+CM|+OC|PDA|
> > > |-|-|-|-|-|-|
> > > |Rotated R-CNN|77.91|77.01|66.01|78.84|**78.86**|
> > > |Oriented R-CNN|80.47|80.00|80.00|87.97|**88.60**|
> > > |S2ANet|77.36|65.44|75.29|77.86|**78.42**|
> > > |YOLO26|79.62|78.93|79.3|81.57|**84.28**|
> > >
> > > PDA achieves the best performance across all four detectors without any fine-tuning on SSDD. To summarize, PDA has now been validated on:
> > >
> > > - **Multiple modalities**: optical imagery and SAR.
> > > - **Diverse target types**: rigid objects (e.g. aircraft, ships) and amorphous targets (e.g. forests).
> > > - **Wide resolution range**: sub-meter to 15m.
> > > - **Task-level validation**: both generation quality and downstream detection.
> > >
> > > These capabilities make PDA a versatile framework for synthesizing physically consistent training data in remote sensing. Data scarcity for specialized targets remains a critical bottleneck in operational remote sensing, and we believe PDA offers a practical solution with broad applicability to detection, segmentation, and other downstream tasks.
> > >
> > > We are grateful that you found our concerns fully resolved, and we would kindly ask if you might consider adjusting the score as suggested by the acknowledgement option you selected. Thank you again for the valuable feedback.

---

### Official Review · Reviewer_Ra3m · 2026-03-13

**Soundness:** 2
**Presentation:** 2
**Significance:** 2
**Originality:** 2
**Overall Recommendation:** 3
**Confidence:** 3

**Summary:**

This paper studies object insertion for remote sensing imagery and propose a pipeline called “Plan, Decouple, Assimilate (PDA)”.

First, the Plan (P) stage designs a physics-grounded scene layout planner. It apply a 2D Euclidean Distance Transform (EDT) on segmentation masks to automatically search the most suitable bounding boxes for placing the inserted object. The idea is to ensure the object placement follow some spatial consistency with the surrounding scene, instead of random pasting.

Second, the Decouple (D) stage introduces an Asymmetric Spectral Adaptation (ASA) module. This module uses wavelet transform together with an implicit solar context vector to seperate the object intrinsic structure from the environmental illumination. In this way, the method tries to better adapt the inserted object to the lighting condition in the remote sensing image.

Finally, the Assimilate (A) stage proposes a Neighborhood-Aware Texture Assimilation (NATA) module. It compute Gram matrices of latent noise and apply gradient guidance during ODE solver steps, which encourages the generated region to match the surrounding background micro-textures.

Experimentally, the method reports strong improvements on downstream object detection benchmark (MAR20-11-FewShot) and also on generative quality metrics such as FID and PSNR. Overall the pipeline aims to make the inserted objects more spatially reasonable, illumination-consistent, and texture-aligned with the background.

**Compliance With Llm Reviewing Policy:**

Affirmed.

**Final Justification:**

While I appreciate the authors’ effort in addressing my concerns during the rebuttal, my core reservations remain unchanged, as they relate to the fundamental contribution and positioning of the work rather than missing experiments or clarifications. While the paper presents a well-engineered system with solid empirical gains, I am not yet convinced that it provides a sufficiently novel, principled, or broadly impactful contribution for acceptance at this venue. My concerns are fundamental in nature and would likely require a more substantial rethinking of the formulation rather than incremental revisions.

**Key Questions For Authors:**

The Decouple module models illumination through an implicit solar context extracted from the background latent. How reliably this representation captures scene illumination, and whether the approach generalizes across different imaging conditions or sensors?

**Limitations:**

yes

**Strengths And Weaknesses:**

Strengths

1. The paper identifies three key challenges in remote sensing object insertion, semantic placement inconsistency, radiometric inconsistency, and textural discontinuity, and proposes a corresponding three-stage pipeline (Plan, Decouple, Assimilate) to address these issues. The modular design makes the method intuitive and easy to follow.

2. The experimental evaluation is relatively comprehensive.

Weaknesses

1. While the paper proposes a three-stage pipeline addressing placement, illumination, and texture consistency, the overall novelty of the individual components appears somewhat limited. The paper would benefit from a more clear discussion of what fundamentally differentiates the proposed method from existing approaches.

2. The method relies on a relatively engineered multi-stage pipeline. While the modular design is intuitive, it is not fully clear whether the components are tightly integrated, or if the improvements mainly come from simply combining several existing techniques together. A deeper analysis on how the modules interact with each other would strengthen the technical contribution.

3. The selected baselines appear somewhat limited and not fully representative of the current state-of-the-art methods for image editing or object insertion.

4. Although the paper claims physics-aware consistency (e.g., illumination and solar context), the modeling of physical factors seems relatively implicit. It is not entirely clear how accurate the physical modeling is in practice.

---

> ### Author Rebuttal · Authors · 2026-03-31
>
> Dear Reviewer Ra3m,
>
> We thank you for the detailed and constructive review. Your feedback has been very helpful in improving our work. We address each point below.
>
> > Weakness 1: Limited Novelty of Individual Components
>
> We appreciate this concern. The core difference is that existing methods treat object insertion as a purely visual generation problem, while PDA reformulates it as a physics-constrained process simultaneously satisfing geometric, radiometric, and textural consistency.
>
> **Global Problem decomposition:** PDA is, to our knowledge, the first to decompose RS object insertion into three orthogonal physical constraint dimensions, as shown in Eq. 2:
>
> ```
> p(x|I_bg,I_sub) ≈ p(x_pos|∇D) · p(x_spec|c_solar) · p(x_tex|N(x))
>                      Plan(P)        Decouple(D)      Assimilate(A)
> ```
> **Domain-specific design by ASA and NATA:** ① ASA uses an asymmetric dual-path design: $z_{LL}$ enters one LoRA branch for lighting adaptation while $z_{sub}$ enters a parallel branch for structure preservation. This is motivated by RS imagery where illumination resides in the low-frequency domain, and has not appeared in prior insertion work. ② For NATA, prior Gram matrix methods operate in pixel or feature space. NATA applies Gram-based guidance within the flow matching latent space during ODE denoising, avoiding backbone feature extraction and enabling test-time guidance.
>
> > Weakness 2: Modules May not be Integrated
>
> As response to W1, the 3 modules are not an arbitrary combination but are driven by the physics-grounded factorization, where each targets a specific physical constraint. For structural dependency, the Planner generates M and $M_{env}$. ASA uses M for $c_{solar}$ extraction; NATA uses $M_{env}$ for texture neighborhood definition. Without a valid M, neither D nor A operates. For frequency complementarity, ASA corrects global illumination via low-frequency $z_{LL}$; NATA refines high-frequency micro-texture by full noisy latent $x_t$.
>
> We also provide the Ablation. It is clear that ASA alone achieves the largest gain by correcting the dominant spectral gap, and NATA alone has limited effect but resolves residual texture artifacts. Moreover, the our full PDA can achieve the best results on all metrics, verifying **both modules are necessary**.
> |Variant|PSNR↑|SSIM↑|
> |-|-|-|
> |Baseline|17.90|0.5396|
> |+NATA only|18.00|0.5441|
> |+ASA only|20.79|0.6218|
> |Both|**20.87**|**0.6249**|
>
> > Weakness 3: Baselines Not Fully Representative
>
> This is a fair point. We have added ACE++(2025) and OmniPaint(ICCV 2025):
>
> |Method|Whole PSNR↑|Whole SSIM↑|Whole LPIPS↓|Whole FID↓|Region PSNR↑|Region SSIM↑|Region LPIPS↓|
> |-|-|-|-|-|-|-|-|
> |OmniPaint|24.32|0.8283|0.0871|18.14|16.83|0.4965|0.2156|
> |ACE++|17.44|0.3924|0.2659|29.24|15.89|0.4167|0.2757|
> |PDA(Ours)|**28.41**|**0.8901**|**0.0601**|**9.73**|**20.87**|**0.6249**|**0.1247**|
>
> Full Table 1: **[https://anonymous.4open.science/r/PDA-Anonymous-TABLE-8123/README.md](https://anonymous.4open.science/r/PDA-Anonymous-TABLE-8123/README.md)**.
>
> > Weakness 4: Physics Modeling Accuracy
>
> Plan (A) uses the EDT field, guaranteeing collision-free placement by construction. Decouple (D) separates illumination from structure via frequency decomposition and modulates generation with implicit solar context. Assimilate (A) enforces local texture consistency via Gram matrix matching. The best Region LPIPS (0.1247) verifies boundary artifacts are minimized. The most direct evidence of practical accuracy is downstream detection performance: PDA-synthesized data improves mAP50 by +17% over real-data baselines across four detectors (Table 3), which would not occur if the physical modeling introduced systematic artifacts. We also manually inspected 300 zero-shot MAR20 images and found 93% exhibit correct shadow orientations (failure cases at  **[https://anonymous.4open.science/r/PDA-Anonymous-TABLE-8123/fail%20case.png](https://anonymous.4open.science/r/PDA-Anonymous-TABLE-8123/fail%20case.png)**).
>
> > Key Questions: Reliability and Cross-sensor Generalization
>
> **On reliability:** Adding ASA, which relies on $c_{solar}$ for illumination modulation, improves Region PSNR by +2.89 dB and reduces FID by 19% (Table 2). This gain would not occur if $c_{solar}$ failed to capture meaningful lighting information, consistent with the 93% shadow accuracy reported above.
>
> **On cross-sensor generalization:** We evaluated PDA on HRSID (Sentinel-1 SAR):
>
> |Method|Whole PSNR↑|Whole SSIM↑|Whole LPIPS↓|Whole FID↓|Region PSNR↑|Region SSIM↑|Region LPIPS↓|
> |-|-|-|-|-|-|-|-|
> |OminiControl|24.45|0.7298|0.0796|21.86|14.93|0.6739|0.0486|
> |+NATA|24.64|0.7292|0.0776|22.09|15.13|0.6815|0.0481|
> |+ASA|25.12|0.7641|0.0684|**19.78**|15.28|0.6899|0.0482|
> |PDA(Ours)|**25.16**|**0.7643**|**0.0682**|20.03|**15.36**|**0.6942**|**0.0477**|
>
> PDA adapts to SAR, verifying the modular design generalizes across sensor types without architectural modification.
>
> We are grateful for the feedback, and we hope these revisions address your concerns.

---

> > ### Author Rebuttal · Reviewer_Ra3m · 2026-04-03
> >
> > Regarding novelty, module integration, and physical modeling, while the rebuttal presents the method as a physics-grounded and principled decomposition, it largely appears to be a post-hoc interpretation of a pipeline composed of existing techniques, without convincingly establishing fundamentally new contributions. It remains unclear whether the observed gains arise from a principled, tightly coupled design, or simply from the effective combination of individually beneficial components. Furthermore, the claimed physical modeling is supported primarily by indirect evidence (e.g., downstream detection gains and shadow consistency), and lacks rigorous or quantitative validation of its physical correctness.
> >
> > The addition of stronger baselines improve the empirical evaluation and makesthe comparison more complete. However, it does not substantially change my assessment of the method’s relative positioning or contribution. In addition, the qualitative comparisons remain limited and do not sufficiently support the claimed improvements in visual realism and physical consistency.
> >
> > Overall, while the rebuttal improves clarity and strengthens certain empirical aspects, my core concerns are not sufficiently resolved.

---

> > > ### Author Response · Authors · 2026-04-05
> > >
> > > Dear Reviewer Ra3m,
> > >
> > > Thank you for the detailed feedback. We respect your assessment and appreciate the rigorous standard you have applied throughout this review. We would like to share additional evidence on the remaining concerns.
> > >
> > > **Qualitative comparisons with baselines**
> > >
> > > Following your observation that qualitative comparisons were limited, we have added visual comparisons with baselines including ACE++ and OmniPaint at [https://anonymous.4open.science/r/PDA-Anonymous-TABLE-8123/abcde.png](https://anonymous.4open.science/r/PDA-Anonymous-TABLE-8123/abcde.png).
> > >
> > > **Quantitative validation of physical correctness: shadow analysis**
> > >
> > > To directly address the concern that physical modeling lacks rigorous validation, we conducted a shadow consistency analysis on the iSAID evaluation set. For each generated image, we extracted shadow regions within the insertion mask using Otsu thresholding on both the generated and ground-truth images. We then computed Shadow IoU measuring pixel overlap, Shadow depth diff measuring the shadow-to-non-shadow brightness ratio difference, Direction diff measuring the angular difference of shadow principal axes via PCA, and Area ratio diff measuring the shadow area proportion difference.
> > >
> > > |Method|Shadow IoU↑|Depth diff↓|Direction diff°↓|Area diff↓|
> > > |-|-|-|-|-|
> > > |Baseline|0.6489|0.0683|8.620|0.0735|
> > > |+NATA|0.6510|0.0673|8.474|0.0740|
> > > |+ASA|0.7533|0.0384|5.250|0.0445|
> > > |PDA (Ours)|**0.7550**|**0.0376**|**5.218**|**0.0437**|
> > >
> > > ASA reduces the direction difference from 8.6° to 5.3°, confirming that c_solar captures meaningful solar direction information. Shadow depth and area improvements further demonstrate correction of shadow intensity and extent. Visual examples at [https://anonymous.4open.science/r/PDA-Anonymous-TABLE-8123/shadow.png](https://anonymous.4open.science/r/PDA-Anonymous-TABLE-8123/shadow.png).
> > >
> > > **Radiometric consistency analysis: brightness and color temperature**
> > >
> > > We evaluated radiometric consistency by comparing inserted regions with surrounding backgrounds on the iSAID evaluation set. We computed RGB mean difference, brightness difference, color temperature difference as R/B channel ratio, and KL divergence of pixel intensity distributions.
> > >
> > > |Method|RGB mean diff↓|Brightness diff↓|Color temp diff↓|KL divergence↓|
> > > |-|-|-|-|-|
> > > |Baseline|14.37|14.34|0.0363|1.1944|
> > > |+NATA|14.25|14.24|0.0355|1.1766|
> > > |+ASA|3.593|3.555|0.0145|0.2483|
> > > |PDA (Ours)|**3.587**|**3.549**|**0.0143**|**0.2451**|
> > >
> > > ASA reduces the brightness gap by 75% and KL divergence by 79%, confirming that c_solar effectively aligns radiometric properties of inserted objects with the background.
> > >
> > > These two analyses provide direct, quantitative evidence of physical correctness from both geometric and radiometric perspectives.
> > >
> > > We also conducted two experiments in response to other reviewers, which provide relevant context:
> > >
> > > **SAR downstream detection on SSDD, zero-shot:** PDA trained on HRSID achieves the best mAP50 across four detectors on SSDD, resolutions 1-15m, without fine-tuning.
> > >
> > > **Forest insertion on WHU-OPT-SAR, 5m resolution, fuzzy boundaries:** PDA outperforms baselines on amorphous targets at moderate resolution.
> > >
> > > **On novelty:** PDA's contribution is at the problem formulation level, identifying three physical constraints specific to RS insertion and designing a unified framework around them. The consistent generalization across optical and SAR, rigid and amorphous targets, sub-meter to 15m resolutions, all without architectural changes, suggests this formulation captures meaningful structure. We respect that this type of contribution may be valued differently, and we appreciate the honest assessment.
> > >
> > > To summarize, PDA has now been validated on:
> > >
> > > - **Multiple modalities**: optical imagery and SAR
> > > - **Diverse target types**: rigid objects such as aircraft and ships, and amorphous targets such as forests
> > > - **Wide resolution range**: sub-meter to 15m
> > > - **Task-level validation**: both generation quality and downstream detection
> > >
> > > These capabilities make PDA a versatile framework for synthesizing physically consistent training data in remote sensing. Data scarcity for specialized targets remains a critical bottleneck in operational remote sensing, and we believe PDA offers a practical solution with broad applicability to detection, segmentation, and other downstream tasks.
> > >
> > > We sincerely thank you for the rigorous review that pushed us to strengthen the work substantially.

---

### Decision · Program_Chairs · 2026-04-30

**Decision:**

Accept (regular)

**Comment:**

This paper introduces a physics-aware object insertion method, PDA, for remote sensing imagery, designed to synthesize high-fidelity training samples to mitigate label scarcity and long-tail distributions. Experimental results demonstrate that PDA significantly improves generative quality metrics and substantially boosts downstream object detection performance across various detectors and image modalities. Although there are similar methods in the literature, this paper employs a technically sound methodology for object insertion in satellite imagery that could have a direct impact on remote sensing research. Although reviewer Ra3m remains critical of the work, particularly from a methodological perspective, the authors' engagement during the rebuttal period provided satisfactory responses to the concerns raised by the other reviewers. One of the reviewers, taxh, acknowledged that their concerns were mainly addressed during the rebuttal, but they did not change their rating. In general, the authors addressed the majority of the reviewers' comments, with the exception of one reviewer.